# Automated Creation of *Digital Cousins* for Robust Policy Learning

**Tianyuan Dai**[1,*], **Josiah Wong**[2,*], **Yunfan Jiang**[1], **Chen Wang**[1], **Cem Gokmen**[1],
**Ruohan Zhang**[1,3], **Jiajun Wu**[1,3], **Li Fei-Fei**[1,3]

[1]Department of Computer Science, Stanford University
[2]Department of Mechanical Engineering, Stanford University
[3]Institute for Human-Centered AI (HAI), Stanford University

**Abstract:** Training robot policies in the real world can be unsafe, costly, and difficult to scale. Simulation serves as an inexpensive and potentially limitless source of training data, but suffers from the semantics and physics disparity between simulated and real-world environments. These discrepancies can be minimized by training in *digital twins*, which serve as virtual replicas of a real scene but are expensive to generate and cannot produce cross-domain generalization. To address these limitations, we propose the concept of ***digital cousins***, a virtual asset or scene that, unlike a *digital twin*, does not explicitly model a real-world counterpart but still exhibits similar geometric and semantic affordances. As a result, *digital cousins* simultaneously reduce the cost of generating an analogous virtual environment while also facilitating better robustness during sim-to-real domain transfer by providing a distribution of similar training scenes. Leveraging digital cousins, we introduce a novel method for their automated creation, and propose a fully automated real-to-sim-to-real pipeline for generating fully interactive scenes and training robot policies that can be deployed zero-shot in the original scene. We find that digital cousin scenes that preserve geometric and semantic affordances can be produced automatically, and can be used to train policies that outperform policies trained on digital twins, achieving 90% vs. 25% success rates under zero-shot sim-to-real transfer. Additional details are available at https://digital-cousins.github.io/.

**Keywords:** Real-to-Sim; Digital Twin; Sim-to-Real Transfer

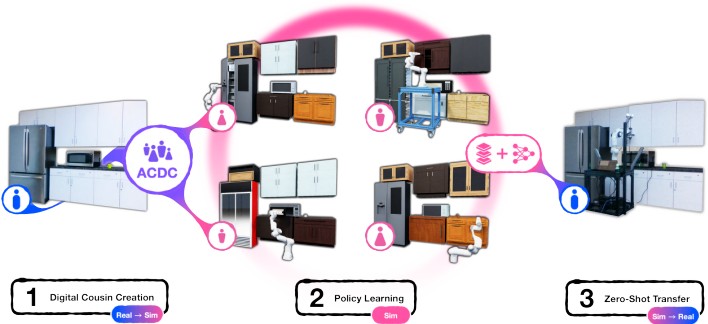

Figure 1: **Overview.** Fully interactive **digital cousin** scenes can be generated completely automatically from a single RGB image. Unlike a digital twin, ***digital cousins*** relax the assumption of completely reconstructing the minute details of a given scene and instead focus on preserving higher-level details, such as spatial relationships and semantic affordances. By leveraging motion planning and ground-truth simulation information, we can automatically collect demonstrations in our digital cousin scenes, augmented with physically plausible randomizations. A policy trained on these synthetic demonstrations can then be deployed zero-shot in the original scene, *without* requiring any additional finetuning.

*Denotes equal contribution. Correspondence to Tianyuan Dai <tydai@stanford.edu>.

8th Conference on Robot Learning (CoRL 2024), Munich, Germany.

# 1 Introduction

Developing and training policy models for robotics in the real world can be unsafe, costly, and difficult to scale with sufficient environment diversity. Learning in simulation is an attractive alternative, as it provides both an inexpensive and potentially limitless source of synthetic data that can be generated at super real-time speed. Unfortunately, policies trained exclusively on simulated data require sim-to-real transfer, and often suffer from the semantics and physics disparity between the simulated and real-world environment. One broad approach to mitigate this issue is to improve policy robustness by augmenting the distribution of synthetic data. Some efforts have sought to randomize over object-centric parameters such as visual semantics [1, 2] or physical parameters [3], whereas other methods have proposed scene-level distributions that are either curated [4–6] or procedurally-generated [7]. These methods, however, can lack the quality of synthetic interaction data at the scale necessary for real-world deployment.

In contrast to generating a distribution of environments, explicitly modeling a fully interactive replica of a specific real-world environment (a *digital twin*) can capture nuanced details within the original environment, but are labor-intensive to generate. While multiple recent efforts have explored reducing this cost by synthesizing real-world scans with either procedural [8, 9] or human-assisted [10] interactive object generation, these approaches can fail to capture necessary affordances needed for downstream tasks and still require human input. Ultimately, digital twins themselves are limited in their scope, as robot policies trained in these environments are optimized for a single real-world instance and cannot generalize to variations in the original scene.

To address the limitations of both extremes of sim-to-real approaches, we first propose the concept of *digital cousins*. We define a *digital cousin* as a virtual asset or scene that, unlike a digital twin, does not explicitly model a real-world counterpart but still exhibits similar geometric and semantic affordances. For example, we would expect an appropriate digital cousin of a real-world cabinet to share a similar layout of handles and drawers, even if the material or detailing differs between the two. A digital cousin of a real-world kitchen might include a similar arrangement of furniture objects, even if individual models slightly differ.

Unlike procedurally generated scenes, digital cousins are fundamentally grounded with respect to a real-world scene, similar to digital twins. However, unlike digital *twins*, digital *cousins* relax the requirement of reconstructing an exact replica, and scenes containing digital cousins instead focus on preserving high-level scene properties, such as spatial object layouts and key semantic and physical affordances. And, as the name suggests, multiple distinct cousins can be generated for a single real-world scene, whereas only a single digital twin can exist for that same scene. Thus, this relaxation serves two purposes: (a) it reduces the need for manual finetuning to guarantee a certain level of fidelity and thereby enables fully automated creation of digital cousins, and (b) it facilitates better robustness to variations in the exact original scene by providing an augmented set of scenes from which to train robot policies.

Leveraging *digital cousins*, we then introduce a novel method for the **A**utomated **C**reation of **D**igital **C**ousins (**ACDC**) that can be used fully-automated end-to-end in a real-to-sim-to-real setup, in which digital cousins generated from a real-world image can be used to train policies deployed zero-shot in the original scene. ACDC leverages DINOv2 [11] as a proxy for measuring similarities between a given real-world asset and candidate digital assets, as it has been shown to visually encode relevant geometric and spatial information from diverse sets of images, and we consider assets with low feature embedding distances as being digital cousins of a given real-world object.

Our contributions are threefold. First, we propose the concept of **digital cousins** and a novel method **ACDC** for their automated creation from a single image requiring zero human input. Second, we provide an automated recipe to train simulation policies in digital cousins. Third, we show that robot manipulation policies trained within digital cousins can match the performance of those trained on digital twins, and can outperform digital twin policies when tested on unseen objects, both in simulation and in the original real-world scene. Code and videos can be found on the project website https://digital-cousins.github.io/.

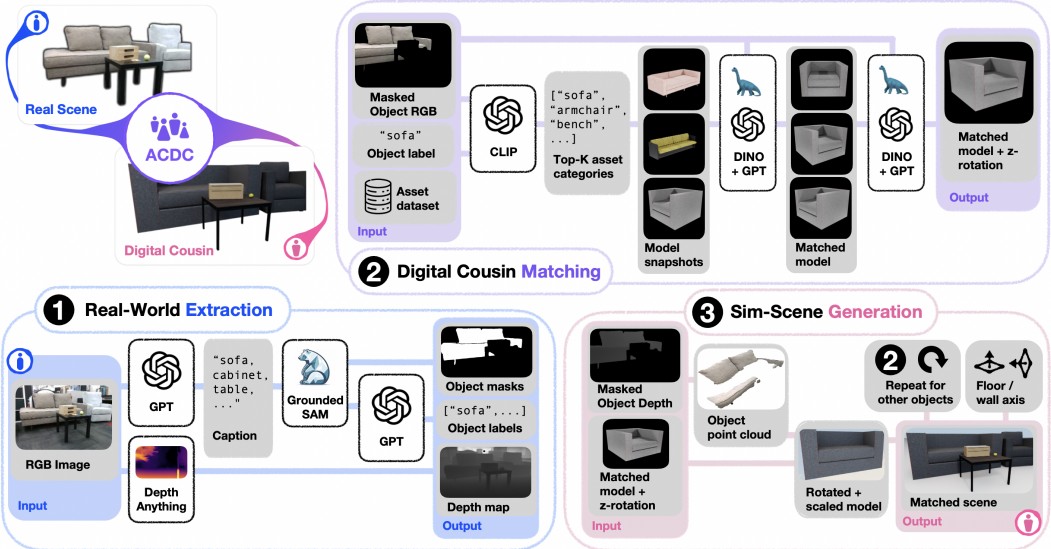

Figure 2: **ACDC Pipeline.** ACDC is composed of three sequential steps. (**1**) First, relevant per-object information is *extracted* the input RGB image. (**2**) Next, we use this information with an asset dataset to *match* digital cousins to each detected input object. (**3**) Finally, we post-process the chosen digital cousins and *generate* a fully-interactive simulated scene.

## 2 Methodology

In this section, we describe our fully automated end-to-end pipeline to generate and leverage digital cousins for sim-to-real policy transfer. In Section 2.1, we describe ACDC, our automated system for generated digital cousins. In Section 2.2, we describe our method for automatically training simulation policies leveraging fully programmatic demonstrations.

### 2.1 Automated Creation of Digital Cousins (ACDC)

ACDC is our automated pipeline for generating fully interactive simulated scenes from a single RGB image, and is broken down into three steps: (**1**) an **extraction** step, in which relevant object masks are extracted from the raw input image, (**2**) a **matching** step, in which we select digital cousins for individual objects extracted from the original scene, and (**3**) a **generation** step, in which the selected digital cousins are post-processed and compiled together to form a fully-interactive, physically-plausible digital cousin scene. An overview of our method can be seen in Fig. 2. Further technical details can be found in Appendix A.

**Real-world extraction.** ACDC only requires a single RGB image $\mathbf{X}$ taken by a calibrated camera with intrinsic matrix $\mathbf{K}$ as the input. To extract individual object masks from the input image, we first prompt GPT-4 [12] to generate captions $\mathbf{c}_j, j \in \{1, ..., M\}$ for all objects observed in $\mathbf{X}$. The captions are then passed to GroundedSAM-v2 [13] with $\mathbf{X}$ to generate a set of detected object masks $\mathbf{m}_i, i \in \{1, ..., N\}$. To re-synchronize the captioning between GroundedSAM-v2 and GPT-4, we re-prompt GPT-4 to select the accurate label $\mathbf{l}_i \in \{\mathbf{c}_j\}_{j=1}^{M}$ for each object mask $\mathbf{m}_i$ from the previously generated caption list.

We additionally require a depth map in order to properly position and rescale matched digital cousins when generating our scene. Depth cameras are widely used but cannot accurately capture reflective surfaces and prevent usage on in-the-wild images. To mitigate these limitations, we leverage Depth-Anything-v2 [14], a state-of-the-art monocular depth estimation model, to estimate the corresponding depth map $\mathbf{D}$ from $\mathbf{X}$. We then extract point cloud $\mathbf{P} = \mathbf{D} \cdot \mathbf{K}^{-1}$, and leverage individual object masks $\mathbf{m}_i$ to generate the subset of points $\mathbf{p_i}$ from $\mathbf{P}$ and pixels $\mathbf{x_i}$ from $\mathbf{X}$ corresponding to that object, resulting in a set of object representations $\{\mathbf{o}_i = (\mathbf{l}_i, \mathbf{m}_i, \mathbf{p}_i, \mathbf{x}_i)\}_{i=1}^{N}$.

**Digital cousin matching.** Given our extracted object representations $\mathbf{o}_i$, we perform a hierarchical search through our virtual asset dataset to match digital cousins. We assume that each asset $i$ in our dataset is assigned a semantically meaningful category $\mathbf{t}_i$, and that each asset model has multiple snapshots $\{\mathbf{i}_{is}\}_{s=1}^{N_{snap}}$ of itself taken under different orientations, including a representative snapshot $\mathbf{I}_i$, forming asset tuples $\{\mathbf{a}_i = (\mathbf{t}_i, \mathbf{I}_i, \{\mathbf{i}_{is}\}_{s=1}^{N_{snap}})\}_{i=1}^{N_{assets}}$, where $N_{assets}$ is the total number of assets included in the dataset. In this work, we use the BEHAVIOR-1K [4] assets, though in practice, our method can use any asset dataset that satisfies the above properties.

For given input object representation $\mathbf{o}_i$, we first select the matching candidate categories by computing the CLIP [15] similarity score between label $\mathbf{l}_i$ and all asset category names $\{\mathbf{t}_i\}_{i=1}^{N_{assets}}$, selecting the top $k_{cat}$ closest categories. Given the selected categories, we then select potential digital cousin candidates amongst all the models within those categories by computing DINOv2 feature embedding distances [11] between the masked object RGB $\mathbf{x}_i$ and representative model snapshots $\mathbf{I}_j$. After selecting $k_{cand}$ candidates, we re-compute the DINOv2 distances over each candidate's individual snapshots $\{\mathbf{i}_{js}\}_{s=1}^{N_{snap}}$ and ultimately select the closest $k_{cous}$ cousins, where each selected cousin consists of a specific virtual asset $\mathbf{A}_c$ and corresponding orientation $\mathbf{q}_c$ based on the selected snapshot.

**Simulated scene generation.** The final step is to compile our matched cousins into a physically plausible digital cousin scene. For given input object information $\mathbf{o}_i$ and corresponding matched cousin information $(\mathbf{A}_c, \mathbf{q}_c)$, we place the asset's bounding box center at the centroid of the corresponding input object point cloud $\mathbf{p}_i$, and then rescale to align with $\mathbf{p}_i$'s extents. We additionally fit floor and wall planes from their obtained point clouds from the **extraction** step, and query GPT-4 to determine whether any objects should be mounted on either the floor or wall. Finally, we de-penetrate all objects so that the scene is physically stable. For additional scene post-processing details, please see Appendix A.3.

## 2.2 Policy Learning

Once we have a set of digital cousins, we train robot policies within these environments that can transfer to additional unseen setups. While our digital cousins are amenable to multiple training paradigms, such as reinforcement learning or imitation learning from humans, we choose to focus on imitation learning from scripted demonstrations, as this paradigm requires no human demonstrations and can instead be coupled end-to-end with our similarly fully autonomous ACDC pipeline.

To facilitate automated demonstration collection in simulation, we implement a set of sample-based skills that leverage both motion planning and ground-truth simulation data. Concretely, our skills include **Open**, **Close**, **Pick**, and **Place**. For specific implementation details, please see Appendix A.4. While currently limited, these skills already enable demonstration collection across a wide range of everyday tasks, such as object rearrangement and furniture articulation.

Moreover, because our generated digital cousin scenes are both modular and configurable, we can easily apply broad domain randomization to these scenes without losing their underlying scene-level semantics through a combination of augmentations, including visual, physics, kinematic (pose and scaling), and instance-level randomization. Using our skills and domain randomization techniques, we can autonomously collect demonstrations across all of our generated digital cousin scenes and train a behavior cloning policy from this offline data. For additional details, see Appendix B.5.

## 3 Experiments

We answer the following research questions through experiments:

> **Q1.** Can ACDC produce high-quality digital cousin scenes? Given a single RGB image, can the recovered digital cousins capture the high-level semantic and spatial details inherent in the original scene?

| Input Scene | Cousin Scene | Scale (m) | Cat. | Mod. | $\mathcal{L}_2$ Dist. (cm) ↓ | Ori. Diff. (rad) ↓ | Bbox IoU ↑ | Cen. IoU ↑ |
|---|---|---|---|---|---|---|---|---|
|  |  | 3.42 | 6/6 | 6/6 | 4.15 ± 2.04 | 0.10 ± 0.14 | 0.64 ± 0.23 | 0.73 ± 0.22 |
|  |  | 4.17 | 8/8 | 8/8 | 7.65 ± 5.62 | 0.05 ± 0.00 | 0.66 ± 0.21 | 0.74 ± 0.16 |
|  |  | 6.89 | 10/10 | 10/10 | 4.77 ± 3.38 | 0.03 ± 0.01 | 0.74 ± 0.20 | 0.77 ± 0.19 |
|  |  | 10.23 | 15/15 | 15/15 | 15.67 ± 8.86 | 0.12 ± 0.11 | 0.59 ± 0.14 | 0.72 ± 0.14 |

Table 1: Quantitative and qualitative evaluation of nearest digital cousin scene reconstruction in a sim-to-sim scenario. 'Scale' is the largest distance between two objects in the input scene. 'Cat.' indicates the ratio of correctly categorized objects to the total number of objects in the scene. 'Mod.' shows the ratio of correctly modeled objects to the total number of objects. '$\mathcal{L}_2$ Dist.' provides the mean and standard deviation of the Euclidean distance between the centers of the bounding boxes in the input and reconstructed scenes. 'Ori. Diff.' represents the mean and standard deviation of the orientation magnitude difference of each centrosymmetric object. 'Bbox IoU' presents the Intersection over Union (IoU) for assets' 3D bounding boxes. 'Cen. IoU' shows the IoU for assets' 3D bounding boxes after aligning their center position. Please refer to Appendix B.1 for more results.

**Q2.** Can policies trained on digital cousins match the performance of policies trained on a digital twin when evaluated on the original setup?

**Q3.** Do policies trained on digital cousins exhibit better robustness compared to policies trained on a digital twin when evaluated on out-of-distribution setups?

**Q4.** Do policies trained on digital cousins enable zero-shot sim-to-real policy transfer?

### 3.1 Digital Cousin Scene Generation via ACDC

**Experiment setup.** We will show quantitative evaluation and qualitative results of recovered digital cousins to answer **Q1**. To quantify the quality of generated digital cousins, we first test our method on a variety of simulated and real scenes to perform both sim-to-sim and real-to-sim digital cousin scene generation, where we input a single RGB image of a simulated scene and generate the closest digital cousins using ACDC. In the sim-to-sim setup, we have guaranteed access to its "digital twin" (i.e., the ground truth category and model), as well as ground truth information about all scene objects' poses and scales, and can quantitatively measure the reconstructive fidelity. In this setting, we measure the proportion of scene objects whose category and model were successfully preserved in the nearest digital cousin to capture the digital cousin's semantic fidelity, and measure the averaged per-object pose error (via $\mathcal{L}_2$ distance and orientation difference) and scale error (via bounding box IoU) to capture its geometric fidelity. However, we do not have access to digital twins for the real-world objects nor ground truth spatial information; instead, we provide qualitative side-by-side comparisons between the real-world scene and its corresponding digital cousin scenes. Sim-to-sim results in Table 1, real-to-sim results in Fig. 3, and additional results in Appendix B.

**Digital cousin generation: Semantic and spatial details are preserved (Q1).** In the sim-to-sim setup, we find that the original per-object category and model are correctly reproduced in most cases. Spatially, we also find that scales and positions of reconstructed digital cousins can similarly match their original counterparts in the input scene. Qualitatively, the side-by-side comparison of our input- and ACDC-generated scenes showcase the immediate visual similarity between the two, and suggest that our quantitative results imply a digital cousin scene quality that can successfully preserve the original scene's object layout. In the real-to-sim setup, we find that ACDC produces reasonable scenes that are both physically plausible and able to preserve scene-level semantic and spatial details.

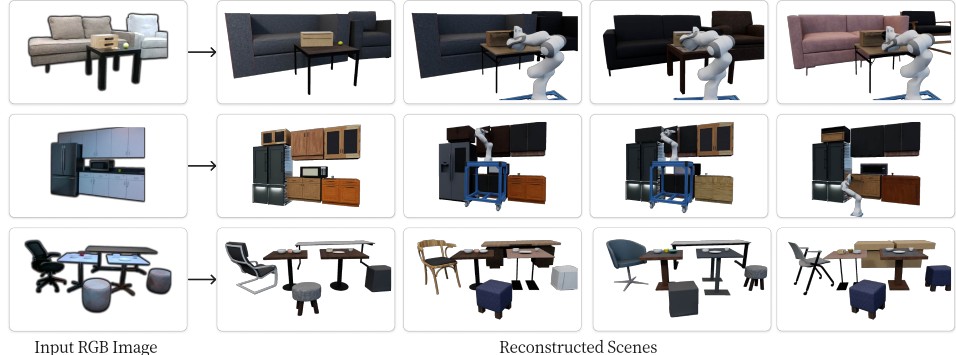

Input RGB Image                                      Reconstructed Scenes

Figure 3: **Qualitative real-to-sim digital cousin scene generation results.** Multiple cousins are shown with a robot collecting demonstrations. Please refer to Appendix B.2 for more results.

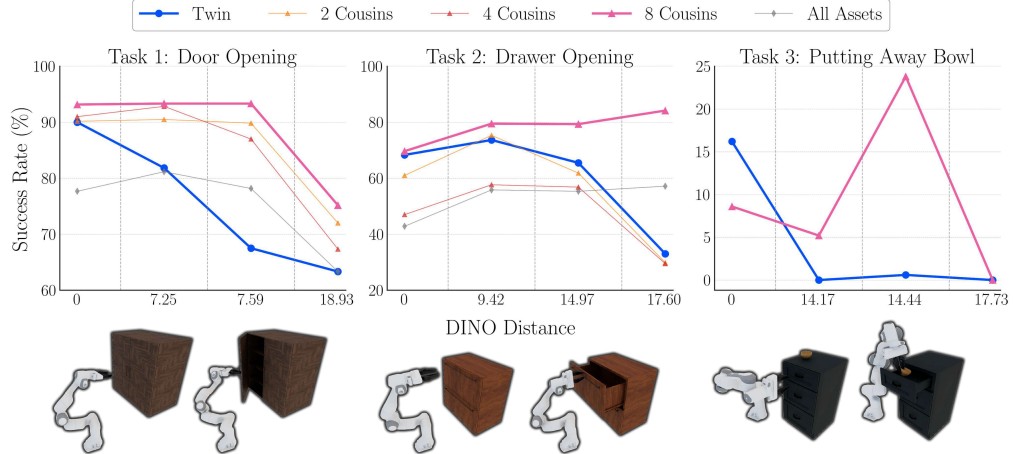

Figure 4: **Sim-to-sim policy results.** Aggregated success rates of policies trained on the exact twin, different numbers of cousins, and all assets in the three nearest categories. Policies are tested on four setups: the exact digital twin, and three increasingly dissimilar setups as measured by DINOv2 embedding distance to probe zero-shot generalization. Note for Task 3, there are much fewer cabinet models that enable the task to be feasible, so we only compare the digital-twin and 8-cousin policies. Note that during digital cousin training data does **not** include any of the evaluation instances. Additional information at Appendix B.6.

**Summary.**   Based on these results, we can safely answer **Q1**: digital cousins can indeed preserve semantic and spatial details of input scenes, reconstructed from a single RGB image that can be accurately positioned and scaled to match the original scene.

### 3.2   Sim-to-Sim Policy Learning with Digital Cousins

**Experiment setup.** To answer **Q2** and **Q3**, we then analyze our ability to train robust robot policies using ACDC-generated digital cousins on three tasks: **Door Opening** and **Drawer Opening**, in which the robot must open furniture equipped with either a revolute- or prismatic-joint, respectively, and **Putting Away Bowl**, in which the robot must open a cabinet's drawer, pick up a bowl on the cabinet and place it in the drawer, and finally close the drawer. We compare policies trained on digital cousins against those trained either exclusively on the digital twin or on all feasible object setups. In each case, our training data consists of 10000 sampled programmatic demonstrations leveraging our analytical skills and divided equally amongst the number of training cabinet instances. However, as the **Putting Away Bowl** task has a much longer horizon compared to the other tasks, we constrain that task's total demonstration count to 2000 to maintain roughly the same training dataset size.

For each policy, we evaluate 100 rollouts over six runs on both the original digital twin setup as well as multiple unseen setups with increasing DINOv2 embedding distance. Our aggregated results are shown in Fig. 4. Additional training details and ablations can be found in Appendix B.6.

**Digital cousin policies can match digital twin policy performance (Q2).** As digital twins perfectly model the target object, policies trained on digital twins serve as oracles for our within-distribution test, and we find that when evaluated on this setup, digital cousin-trained policies can often perform similarly to its equivalent digital twin policy despite not being trained on that specific setup. We hypothesize that because our digital cousin policies are trained on data collected across different setups, it can cover a broad state space that generalizes well to the original digital twin setup. However, on the other extreme, we also find that policies trained on all feasible assets perform much worse compared to the digital twin policy, suggesting that naive domain randomization is not always unequivocally useful and that digital cousins serve as a more beneficial, *conditional* form of randomization.

**Digital cousins improve policy robustness (Q3).** In held-out setups unseen by both the digital twin and digital cousin policies, we find that the performance disparity sharply increases. While policies trained on digital cousins exhibit more robust performance across these setups, the digital twin policy exhibits significant degradation. This suggests that digital cousins can improve policy robustness to setups that are unseen but still within the distribution of cousins that the policy was trained on. Moreover, policies trained on all assets exhibit consistent but low performance, again highlighting the improvement of guided domain randomization via digital cousins.

**Digital cousins provide a proxy for out-of-distribution performance (Q3).** We additionally observe that digital twin policy performance generally degrades proportionally as the DINOv2 embedding distance increases across evaluation setups. This suggests that digital cousins may serve as a proxy for out-of-distribution performance, with "further away" setups capturing setups that are proportionally further away from the data distribution seen in the original setup.

### 3.3 Sim-to-Real Policy Learning with Digital Cousins

Ultimately, we want our pipeline to accelerate sim-to-real policy transfer, where digital cousins may cover a conditioned but wider distribution to mitigate the sim-to-real gap. To evaluate our approach, we use a real-world IKEA cabinet and its corresponding digital twin model, train both a digital cousin policy using cousins matched from ACDC and multiple digital twin policy baselines using the virtual asset, and then evaluate zero-shot on the real cabinet. Our results are shown in Fig. 5.

**Digital cousins can enable zero-shot sim-to-real policy transfer (Q4).** We find that while both the digital twin and digital cousin policies perform well in simulation, only the digital cousin policy is able to transfer to the real world. We hypothesize that because digital cousins provide a wider distribution of training data, the resulting sim policy is better able to overcome the sim-to-real domain gap resulting from asset modeling and sensor perception errors. Moreover, we find that naive domain randomization alone is insufficient to overcome the sim-to-real domain gap, and that leveraging digital cousins can better overcome this gap and reduce the need for exact twin reconstruction.

### 3.4 Real-to-Sim-to-Real Scene Generation and Policy Learning

Finally, we test our full pipeline and automated policy learning framework end-to-end with a fully in-the-wild kitchen scene. We find that our policy can successfully open the kitchen cabinet after being trained exclusively in simulation on digital cousins, as seen in Fig. 1. Experiment videos and additional results can be found at https://digital-cousins.github.io/.

**Summary.** Based on these results, we can safely answer **Q2**, **Q3**, and **Q4**: Policies trained using digital cousins exhibit comparable in-distribution and more robust out-of-distribution performance compared to policies trained on digital twins, and can enable zero-shot sim-to-real policy transfer.

## 4  Related Work

**Real-to-Sim Scene Creation for Robotics**
Creating realistic and diverse digital assets and
scenes from real-world inputs is a prevalent and
long-standing problem [16–19]. Within robot
learning, real-to-sim scene creation has been
achieved through manual curation [4, 10, 20–
25], procedural generation [26–28], few-shot
interactions [8, 29, 30], inverse graphics [31],
and more recently foundation model-assisted
generation [32, 33]. However, these methods
either cannot handle scene-level generation,
require human labor, or cannot retain physical
plausibility. In contrast, ACDC is fully auto-
mated and the recovered digital cousins are
faithful to the input physical scenes.

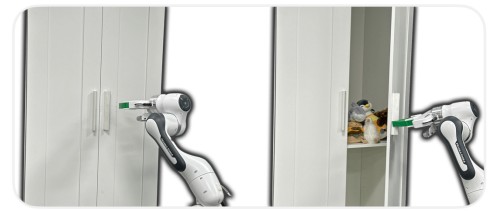

| Policy | Sim Success | Real Success |
|---|---|---|
| Twin | 100% | 25% |
| Twin + ↑ DR | 70% | 55% |
| Twin + Cousin | 92% | 95% |
| Cousin | 94% | 90% |

Figure 5: **Zero-shot real-world evaluation of digi-
tal cousin policy vs. digital twin baselines.** Task is
**Door Opening** on an IKEA cabinet. Metric is success
rate: sim/real results averaged over 50/20 trials. Twin
+ ↑DR is trained using increased domain (pose, scale)
randomization, and Twin + Cousin is trained on both
twin and cousin data.

**Policy Learning with Synthetic Data**   Data
synthesis for robot learning can alleviate the
burden of collecting data in the real world with
physical robots [34–36]. To synthesize complete robotic trajectories (sequences of observation-
action pairs), researchers develop action primitives operating on privileged information available
in simulation [37–40], leverage task and motion planning (TAMP) [41] to generate robot mo-
tions [31, 42, 43], train and distill RL policies [44–46], and automate data generation given an initial
set of human demonstrations [33, 47, 48]. In this vein, our work also leverages primitive skills
for efficient and robust data collection. However, unlike previous methods, which use generative
models to synthesize data [49, 50], our reconstructed scenes are physically plausible, which eases
policy learning and better facilitates transfer to real hardware.

**Sim-to-Real Policy Transfer**   Seamlessly deploying robot policies learned in the simulation to the
real world is critical. Successful sim-to-real transfer has been demonstrated on dexterous in-hand
manipulation [45, 46, 51–53], robotic-arm manipulation [54–64], quadruped locomotion [65–68],
biped locomotion [69–74], and quadrotor flight [75, 76]. Methods to bridge sim-to-real gaps
mainly include domain randomization [51, 77–79], system identification [60, 65, 80, 81] and
simulator augmentation [82–84]. Notably, recent work demonstrates robust real-world deployment
of manipulation policies by training on diverse simulated scenes [10, 31]. Our work expands
the simulation training coverage and hence further robustifies policies by training on "digital
cousins"—a wider distribution than the nearest-asset training scenario.

## 5  Conclusion

Digital cousins can be quickly generated by a fully automated pipeline, called ACDC, from a single
real-world RGB image. We find that policies trained on these digital cousins are more robust than
those trained on digital twins, with comparable in-domain performance and superior out-of-domain
generalization, and enable zero-shot sim-to-real policy transfer.

**Limitations.** Our system has a few limitations. First, ACDC is bounded by the diversity of its
underlying asset dataset. While BEHAVIOR-1K contains thousands of unique assets, we find that it
is still insufficient to densely capture the real-world distribution of objects. Second, because ACDC
is built upon multiple large pretrained models, our pipeline inherits the limitations of these models,
including adversarial and out-of-distribution scenes. Third, policy learning with digital cousins can
still be significantly improved and can benefit from recent advancements in robot learning, such as
diffusion policies [85].

**Acknowledgments**

We are grateful to the SVL PAIR group for helpful feedback and insightful discussions. This work is in part supported by the Stanford Institute for Human-Centered AI (HAI), ONR MURI N00014-22-1-2740, ONR YIP N00014-24-1-2117, ONR MURI N00014-21-1-2801, and Schmidt Sciences. Ruohan Zhang is partially supported by the Wu Tsai Human Performance Alliance Fellowship.

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

# Appendix

## A  Additional Cousin Creation Details

### A.1  Offline Dataset Generation

Cousins creation requires a large-scale asset set. We adopt BEHAVIOR-1K [4], which includes over 10,000 object assets. The goal of this stage is to preprocess the whole asset set for later usage. Since objects may have occlusion in the input image, common approaches that can estimate the scale and orientation of real objects such as point cloud registration [86, 87] and monocular pose estimation methods [88] are not feasible because these generally require two complete, unobstructed point clouds for a given object. Instead, we choose to represent each asset as a set of visual 2D images, under the expectation that we will use a visual encoder (such as DINOv2) downstream to match geometric correspondences between objects. For our given dataset, we rotate each asset $\mathbf{a}_i$ in the whole asset set and take snapshots from a fixed camera pose $\mathbf{P}_{sim}$, resulting in a set of images $\{\mathbf{i}_{is}\}_{s=1}^{N_{snap}}$ and representative snapshot $\mathbf{I}_i$. Each asset $\mathbf{a}_i$ is pre-annotated with its own semantically-meaningful category $\mathbf{t}_i$. This results in asset tuples $\{\mathbf{a}_i = (\mathbf{t}_i, \mathbf{I}_i, \{\mathbf{i}_{is}\}_{s=1}^{N_{snap}})\}_{i=1}^{N_{assets}}$, where $N_{assets}$ is the total number of assets included in the BEHAVIOR-1K dataset. Note that this stage occurs once offline, and can be cached when running ACDC.

### A.2  Mounting Type

We observe that scene objects often serve different semantic roles and fall under difference pose distributions depending on whether an object is fixed with respect to the room. Therefore, as mentioned in Section 2.1, we leverage this inductive bias and prompt GPT to determine if an object is mounted on a wall or not. This distinction helps address a key limitation with our one-shot approach: because of heavy occlusion resulting from a single camera view, objects such as televisions or cabinets that are mounted to walls may only have its frontal face observed from a single camera view, resulting in a insufficient extracted point cloud that does not fully capture its underlying volumetric depth. television or a cabinet fixed on a wall, a frontal view image may only cover the frontal face of the mounted object.

We mitigate this limitation by prompting GPT to classify each object into one of three semantic categories: (1) **Wall Mounted**: An object is fixed on a wall with nothing closely beneath it; (2) **On Floor or On Another Object**: An object is placed on the floor, or on another object, but the object does not touch a wall; (3) **Mixture**: An object is not mounted on a wall, but one of its face touches a wall, like a bookshelf putting on the floor but touches the wall behind it, or a microwave oven putting on a cabinet but its back face touches the wall behind it. In cases (1) and (3), we also require GPT to specify the specific wall on which the object is mounted by feeding all masked walls in the input image generated by Grounded-SAM-v2. In practice, we first prompt GPT to identify whether an object is installed or fixed on one or more walls and to specify which wall(s) it is attached to. If the object is mounted on a wall, it is classified as (1) **Wall Mounted**. For objects not installed on any wall, we prompt GPT to determine if the object is aligned with and in contact with one or more walls. This step further classifies the object into either mounting type (2) or (3). Users have the option to disable the second prompt, thereby distinguishing only whether an object is wall-mounted or not. Please see Appendix A.3 for how objects with different mounting types are processed.

### A.3  Generated Scene Post-Processing

After putting all assets in the correct position in the **Simulated scene generation** stage described in Section 2.1, we post-process each asset for a physically plausible scene. For each asset $i$, we should have its bounding center position $\mathbf{p}_i^{cen}$, bounding box's top-right vertex position $\mathbf{p}_i^{TR}$, and bounding box's bottom-left vertex position $\mathbf{p}_i^{BL}$. First, we sort all assets from low to high by sorting $\mathbf{p}_i^{cen}$ in ascending order, and project each asset's 3D bounding box to the x-y plane, resulting in a 2D polygon $poly_i$ for each asset $i$. We then infer "on top" relationships from our sorted asset list.

For each asset $i$, we search over all assets with lower $\mathbf{p}_i^{cen}$ to determine another asset $j$ right beneath it. Whenever the overlapped area between the lower asset $j$'s projected 2D polygon $poly_j$ and the current asset's projected 2D polygon $poly_i$ exceeds $70\%$ of the area of either one of the 2D polygons, i.e., $area(intersect(poly_i, poly_j)) > 0.7 \cdot \min(area(poly_i), area(poly_j))$, we determine that the higher asset $i$ is on top of the lower asset $j$, and the lower asset $j$ is beneath the higher asset $i$. Intuitively, this checks for vertical spatial alignment between two objects. If no matching asset is found, the asset is regarded as being on top of the floor. After all assets have been evaluated in this way, each asset should have another asset or floor beneath it after performing the above searching.

Next, we post-process all assets based on their mounting type: For an asset $i$ with mounting type (1) (Wall Mounted), we first adjust its scale and orientation, and then adjust its position. Since asset $i$ is mounted on a wall, we determine the face of asset $i$ that should be adjusted such that it becomes parallel to the wall. First, we fit a plane to the wall from its corresponding extracted point cloud. We then compute the minimum rotation that aligns either the object's local x or y axis with the normal vector of the wall plane. Finally, we compute the distance between $\mathbf{p}_i^{cen}$ and the wall, and rescale and translate asset $i$ in the x-y plane such that the object's rear face is co-planar with the wall plane and object's front face maintains its same position. Finally, we de-penetrate this object from others by adjusting $\mathbf{p}_i^{cen}$'s z value: We increase $\mathbf{p}_i^{cen}$ by $z(\mathbf{p}_j^{TR}) - z(\mathbf{p}_i^{BL})$, if $z(\mathbf{p}_j^{TR}) > z(\mathbf{p}_i^{BL})$, where $z(\cdot)$ if the z coordinate of a 3D vector, and $j$ is the index of the asset beneath asset $i$, and then fix asset $i$ on the wall that GPT selected for asset $i$. When $z(\mathbf{p}_j^{TR}) \leq z(\mathbf{p}_i^{BL})$, we directly fix asset $i$ on the wall without adjusting its position.

For an asset $i$ with mounting type (2) (On Floor or On Another Object), we similarly de-penetrate by placing asset $i$ on top of asset $j$ by adjusting $\mathbf{p}_i^{cen}$ by $|z(\mathbf{p}_j^{TR}) - z(\mathbf{p}_i^{BL})|$. For an asset $i$ with mounting type (3) (Mixture), we adjust the orientation and scale in the same way as assets with mounting type (1), and then adjust $\mathbf{p}_i^{cen}$ in the same way as assets with mounting type (2).

Finally, we check for collisions between the collision meshes of each pair of placed assets and adjust their positions in the x-y plane to avoid any overlap.

## A.4 Skill Definition

In order to bootstrap automated demonstration collection, we define a library of analytical and sampling-based skills that can be chained together to solve long-horizon tasks, such as the **Putting Away Bowl** task. For collision-free motion planning, we leverage CuRobo [89]. For sampling-based grasp generation, we leverage Grasp Pose Generator (GPG) [90] [91] based on a given object's sampled point cloud from its analytical mesh. Below, we briefly describe the high-level implementation of each skill:

**Open.** This skill consists of five steps: **Approach**, which computes a collision-free trajectory towards a point offset in front of the desired handle to articulate, **Converge**, which computes an open-loop straight-line trajectory to the actual grasping point on the handle, **Grasp**, which closes the gripper to grasp the handle, **Articulate**, which computes an open-loop analytical trajectory to articulate the link, and **Ungrasp**, which opens the gripper to release the handle.

For a given articulated object, we leverage ground-truth knowledge of its geometric affordances to compute a corresponding trajectory. Given a specific articulated asset **a** and desired link to articulate **l**, we first infer the link's corresponding handle location by shooting rays towards the link and define the mean handle location as mean location over the rays with the shortest distance. This assumes that the most protruding geometric feature corresponds to the handle. Given handle location, we inspect **l**'s parent link **j**'s properties, determining its type (prismatic or revolute) and pose with respect to the handle. Given this information, we can compute a desired analytical trajectory for the handle to open link **l**. This can easily be transformed into the robot frame, and offset according to the robot's end-effector size.

**Close.** This implementation is nearly identical to **Open**, though for computing the desired articulation trajectory, the start / end points are reversed.

**Pick.** This skill consists of three steps: **Move**, which computes a collision-free trajectory towards a sampled grasping point, **Grasp**, which closes the gripper to grasp the object, and **Lift**, which computes an open-loop trajectory to lift the object slightly.

Note that during the **Move** phase, we sample grasping points that are both feasible, collision-free, and minimize robot gripper orientation changes to avoid bad robot configurations.

**Place.** This skill consists of three steps: **Move**, which computes a collision-free trajectory towards a sampled placement pose, **Ungrasp**, which opens the gripper to release the object, and **Lift**, which computes an open-loop trajectory to lift the gripper slightly.

This skill assumes that an object is already grasped prior to its execution. We assume the desired placement pose is a kinematic predicate relative to another scene object, e.g.: `inside(cabinet)`. Given this predicate, we use rejection sampling to sample collision-free poses for the robot's end-effector and grasped object that satisfy the given predicate, prioritizing poses that minimize end-effector rotation.

### A.5  Demonstration Collection

We use fully automated demonstrations using our programmatic skills defined above. For the **Door Opening** and **Drawer Opening** tasks, this simply consists of executing the **Open** skill. For the **Putting Away Bowl** task, this consists of a **Open**, **Pick**, **Place**, **Close** sequence. We use rejection sampling so that our resulting dataset only includes successes, that is, if any skill execution fails midway, we do not save that episode. This allows us to significantly increase the randomization range between episodes without being limited by poor edge cases.

Across all tasks, we randomize the agent's pose as well as scene objects' poses and scales between episodes.

### A.6  Using DINOv2 for Digital Cousin Matching

For a given input image $\mathbf{x}$ and set of candidate matching images $\{\mathbf{i}_j\}_{j=1}^N$, we define the top-1 matched candidate through a DINOv2-based voting system. First, we pass both input image $\mathbf{x}$ and all candidate images $\{\mathbf{i}_j\}_{j=1}^N$ through DINOv2, retrieving their feature patches $\mathbf{e}$ and $\{\mathbf{f}_j\}_{j=1}^N$, respectively. Next, we compute the nearest neighbor (defined as the L2-norm) in the DINOv2 feature embedding space for each pixel in $\mathbf{e}$ over all pixels across all candidate feature embeddings $\{\mathbf{f}_j\}_{j=1}^N$, and record the running count of nearest neighbors across all candidates $j \in \{1, ..., N\}$. The top-1 matched candidate is then the candidate with the highest count of per-pixel nearest neighbors – i.e.: the candidate image $\mathbf{i}_j$ that has the highest number of closest visual feature correspondences to input image $\mathbf{x}$. For top-k matched candidates, we repeat the process iteratively, selecting the top-1 each time and subsequently removing the selected $\mathbf{i}_j$ during proceeding iterations. We leverage GPU-accelerated nearest neighbor computations using the open-source faiss [92] package.

Given a matched pair of images $\mathbf{x}$, $\mathbf{i}_j$, we define the DINOv2 embedding distance as the average nearest neighbor L2-distance between each pixel in corresponding input feature map $\mathbf{e}$ and all pixels in corresponding matched feature map $\mathbf{f}_j$. Note that we exclude the largest 10% of nearest neighbor distances in this calculation, as we find empirically that the sorted results across matched candidates are more salient with these outliers removed.

### A.7  Additional Real-to-Sim Details

In this subsection, we provide additional implementation details of ACDC real-to-sim pipeline:

**Depth image and point cloud processing.** One key design decision is to use synthetic depth via Depth-Anything-v2 [14], instead of a dedicated depth camera. This decision is guided by our observation that it performs more consistently on reflective surfaces. However, this synthetic depth approach still generates artifacts occurring near object boundaries, the image periphery, and under lighting changes. To further remove noise in object point clouds, we apply DBSCAN clustering [93] on each object point cloud $\mathbf{p}_i$ to filter out noisy points.

**Orientation Refinement.** DINO performs a rough estimation of asset orientations, which for most objects the orientation is sufficiently accurate. However, we additionally provide an option to further refine the orientation refinement based on an object's extracted point cloud. By computing the z-aligned minimum bounding box of the given point cloud, we can apply an additional z-rotation to DINO's outputted estimated orientation so that the matched asset's canonical xy-axes aligns with the computed minimum bounding box frame. We find this is especially useful for object's that have sharp geometric boundaries, such as furniture objects.

**Heuristics for articulated objects.** In this project, articulated objects refer to those with doors (revolute) and drawers (prismatic). To ensure the selected digital cousins of an articulated object are also articulated, so that door opening or drawer opening demos can be collected on all digital cousins, we propose to search digital cousins for articulated objects only among articulated assets. Because we have ground-truth information for all of our dataset assets, we know *apriori* which assets are articulated. During the **Real-world extraction** stage, we additionally prompt GPT to determine whether objects are articulated.

An optional heuristics is to apply a door/drawer count threshold on digital cousin creation of articulated objects. During the **Offline Dataset Generation** stage, we can count the number of doors (revolute joints) and drawers (prismatic joints). When creating cousins, we only search among assets with "similar" number of drawers and doors. This threshold is open to users to set. In all of our real-to-sim results, we set the threshold to 2 in the nearest cousin selection too guarantee affordance preservation, but do not apply this heuristic to the rest of the scenes.

**GPT API Usage.** We use GPT-4o for the real-to-sim pipeline.

**Inference Time.** While ACDC 's overall wall-clock time varies as a function of scene complexity, in general, we empirically observe the following:

> **Step 1.** [Real-World Extraction] takes around 7 seconds per object.
>
> **Step 2.** [Digital Cousin Matching] takes around 20 seconds to select one digital cousin for an object.
>
> **Step 3.** [Simulated Scene Generation] takes less than 30 seconds for a whole scene.

# B  Additional Experimental Details

## B.1  Visual Encoder Ablation Study

In this subsection, we extend Section 3.1 of our main paper by conducting an ablation study on the real-to-sim pipeline in a sim-to-sim setting. We seek to evaluate whether DINO is sufficient for digital cousin matching, or if applying GPT to finetune DINO's selections can result in improved performance. Our quantitative and qualitative results cover the following comparisons: **(a)** DINO Model Selection & GPT Orientation Selection; **(b)** DINO Model Selection & DINO Orientation Selection; **(c)** GPT Model Selection & GPT Orientation Selection; **(d)** GPT Model Selection & DINO Orientation Selection.

**DINO Model Selection** involves selecting an asset $\mathbf{A}_c$ as the best digital cousin of an object based solely on the DINOv2 embedding distances between the masked object RGB $\mathbf{x}_i$ and all assets' representative model snapshots $\mathbf{I}_j$ within the nearest $k_{cat}$ categories. While DINO Model Selection

| Input Scene | Cousin Scene | Scale (m) | | Cat. | Mod. | L2 Dist (cm) ↓ | Ori. Diff. ↓ | Bbox IoU ↑ | Ori. Bbox IoU ↑ |
|---|---|---|---|---|---|---|---|---|---|
| | | 3.68 | (a) | 5/5 | 5/5 | 5.27 ± 2.85 | 0.07 ± 0.07 | 0.75 ± 0.14 | 0.64 ± 0.07 |
| | | | (b) | | | 5.27 ± 2.85 | 0.07 ± 0.07 | 0.75 ± 0.14 | 0.64 ± 0.07 |
| | | | (c) | 5/5 | 5/5 | 5.27 ± 2.85 | 0.07 ± 0.07 | 0.75 ± 0.14 | 0.64 ± 0.07 |
| | | | (d) | | | 5.27 ± 2.85 | 0.07 ± 0.07 | 0.75 ± 0.14 | 0.64 ± 0.07 |
| | | 3.42 | (a) | 6/6 | 4/6 | 4.79 ± 1.52 | **0.07 ± 0.03** | 0.54 ± 0.32 | 0.52 ± 0.28 |
| | | | (b) | | | 4.85 ± 1.52 | 0.08 ± 0.01 | 0.54 ± 0.32 | 0.52 ± 0.28 |
| | | | (c) | 6/6 | **6/6** | **4.15 ± 2.04** | 0.10 ± 0.14 | 0.64 ± 0.23 | 0.73 ± 0.22 |
| | | | (d) | | | 5.03 ± 1.53 | **0.10 ± 0.11** | 0.54 ± 0.32 | 0.52 ± 0.29 |
| | | 2.91 | (a) | 6/6 | 4/6 | **5.51 ± 2.71** | 0.03 ± 0.00 | 0.64 ± 0.16 | **0.60 ± 0.17** |
| | | | (b) | | | 5.97 ± 2.56 | 0.03 ± 0.00 | 0.64 ± 0.16 | **0.60 ± 0.17** |
| | | | (c) | 6/6 | **5/6** | 7.13 ± 4.77 | 0.16 ± 0.19 | 0.54 ± 0.24 | 0.51 ± 0.21 |
| | | | (d) | | | 5.60 ± 2.92 | **0.10 ± 0.10** | **0.65 ± 0.18** | **0.60 ± 0.17** |
| | | 3.54 | (a) | 5/5 | 2/5 | **6.47 ± 2.79** | 0.04 ± 0.01 | **0.64 ± 0.23** | **0.65 ± 0.28** |
| | | | (b) | | | 6.83 ± 3.20 | **0.03 ± 0.01** | 0.63 ± 0.24 | 0.64 ± 0.30 |
| | | | (c) | 5/5 | **3/5** | 6.51 ± 2.77 | 0.03 ± 0.01 | **0.64 ± 0.22** | 0.60 ± 0.20 |
| | | | (d) | | | 6.90 ± 3.21 | 0.03 ± 0.01 | 0.62 ± 0.24 | 0.58 ± 0.22 |
| | | 3.24 | (a) | 5/6 | 3/6 | 6.64 ± 3.34 | 0.24 ± 0.20 | 0.57 ± 0.21 | 0.58 ± 0.15 |
| | | | (b) | | | 5.92 ± 3.32 | **0.06 ± 0.03** | **0.69 ± 0.14** | 0.66 ± 0.14 |
| | | | (c) | 5/6 | **5/6** | 6.00 ± 3.79 | 0.06 ± 0.05 | 0.68 ± 0.14 | 0.65 ± 0.14 |
| | | | (d) | | | **5.33 ± 3.77** | **0.04 ± 0.03** | **0.69 ± 0.15** | **0.67 ± 0.15** |
| | | 4.17 | (a) | 8/8 | 3/8 | 7.81 ± 4.87 | 0.05 ± 0.00 | 0.65 ± 0.18 | 0.64 ± 0.17 |
| | | | (b) | | | 7.91 ± 5.04 | 0.05 ± 0.00 | 0.65 ± 0.19 | 0.64 ± 0.16 |
| | | | (c) | 8/8 | **8/8** | **7.65 ± 5.62** | 0.05 ± 0.00 | **0.66 ± 0.21** | **0.74 ± 0.16** |
| | | | (d) | | | 7.94 ± 5.01 | 0.05 ± 0.00 | **0.66 ± 0.17** | 0.63 ± 0.17 |
| | | 6.89 | (a) | 10/10 | 6/10 | 7.27 ± 5.51 | 0.46 ± 0.50 | 0.54 ± 0.30 | 0.55 ± 0.25 |
| | | | (b) | | | 6.31 ± 4.78 | **0.24 ± 0.49** | 0.64 ± 0.26 | 0.62 ± 0.25 |
| | | | (c) | 10/10 | **10/10** | **4.77 ± 3.38** | **0.03 ± 0.01** | **0.74 ± 0.20** | **0.77 ± 0.19** |
| | | | (d) | | | 5.96 ± 4.35 | 0.05 ± 0.04 | 0.70 ± 0.18 | 0.63 ± 0.18 |
| | | 10.23 | (a) | 15/15 | 12/15 | 17.40 ± 9.05 | 0.17 ± 0.17 | 0.49 ± 0.20 | 0.50 ± 0.19 |
| | | | (b) | | | 17.01 ± 9.34 | **0.15 ± 0.14** | 0.51 ± 0.20 | 0.50 ± 0.20 |
| | | | (c) | 15/15 | **15/15** | **15.67 ± 8.86** | 0.12 ± 0.11 | **0.59 ± 0.14** | **0.72 ± 0.14** |
| | | | (d) | | | 17.09 ± 9.26 | **0.11 ± 0.10** | 0.55 ± 0.21 | 0.53 ± 0.21 |

Table 2: Quantitative evaluation of nearest digital cousin scene reconstruction in a sim-to-sim scenario. This table is an extension of Table 1 in the main paper. 'Cat.' indicates the ratio of correctly categorized objects to the total number of objects in the scene. 'Mod.' shows the ratio of correctly modeled objects to the total number of objects in the scene. 'L2 Dist' provides the mean and standard deviation of the Euclidean distance between the centers of the bounding boxes in the input and reconstructed scenes. 'Ori. Diff.' represents the mean and standard deviation of the orientation magnitude difference of each non-uniformly symmetric object. 'Bbox IoU' presents the Intersection over Union (IoU) for axis-aligned 3D bounding boxes. 'Ori. Bbox IoU' displays the IoU for oriented 3D bounding boxes.

| Method | Cousin Rank | Training Models | | | |
|---|---|---|---|---|---|
| | | Twin | 2 Cousins | 4 Cousins | 8 Cousins |
| DINO | 0 | 97 94 93 93 92 87 | 93 93 93 92 88 85 | 92 91 90 90 88 85 | 90 89 88 88 87 87 |
| | 2 | 88 85 94 88 90 81 | 94 94 89 95 90 89 | 91 97 94 89 96 88 | 92 89 90 89 93 90 |
| | 6 | 37 42 60 38 45 74 | 87 91 87 92 93 93 | 90 93 92 91 96 95 | 90 86 94 88 95 90 |
| | OOD | 30 62 58 51 38 48 | 65 64 63 44 46 44 | 61 52 61 74 55 68 | 72 54 49 61 50 55 |
| DINO + GPT | 0 | 97 94 93 93 92 87 | 98 94 93 91 89 86 | 92 92 92 90 88 87 | 94 92 91 88 86 79 |
| | 2 | 88 85 94 88 90 81 | 95 94 95 92 93 95 | 95 92 88 95 97 96 | 91 95 90 97 88 92 |
| | 6 | 88 87 90 91 91 87 | 87 95 92 94 93 91 | 87 92 86 94 85 91 | 88 88 88 86 87 89 |
| | OOD | 30 62 58 51 38 48 | 44 43 57 42 46 55 | 56 43 56 63 47 54 | 51 37 51 43 64 52 |
| CLIP | 0 | 97 94 93 93 92 87 | 85 82 74 71 61 58 | 65 63 62 55 54 52 | 84 82 81 76 72 70 |
| | 2 | 80 80 87 80 74 89 | 97 97 92 90 90 89 | 87 76 87 79 85 81 | 88 87 86 90 83 81 |
| | 6 | 61 67 77 68 64 82 | 89 94 91 99 96 90 | 84 89 90 83 90 90 | 90 90 90 85 78 82 |
| | OOD | 30 62 58 51 38 48 | 44 46 51 52 74 29 | 65 63 85 58 75 67 | 64 62 64 57 56 64 |
| CLIP + GPT | 0 | 97 94 93 93 92 87 | 90 87 86 84 82 81 | 93 91 90 90 89 85 | 91 88 84 84 83 82 |
| | 2 | 1 0 1 2 1 0 | 0 2 1 1 0 0 | 0 0 0 0 1 0 | 2 1 1 0 0 2 |
| | 6 | 88 85 94 88 90 81 | 93 91 86 91 92 90 | 96 93 95 95 88 95 | 91 95 90 82 96 93 |
| | OOD | 30 62 58 51 38 48 | 48 69 63 53 70 62 | 61 62 62 49 34 49 | 55 50 62 71 46 68 |

Table 3: Success rates (%) of all policies used in Fig. 7. "Cousin Rank" shows the rank of test cousins selected by each method. Notice that all test assets are not seen during policy training. "OOD" stands for an asset that is not selected as top-12 digital cousin by all four methods.

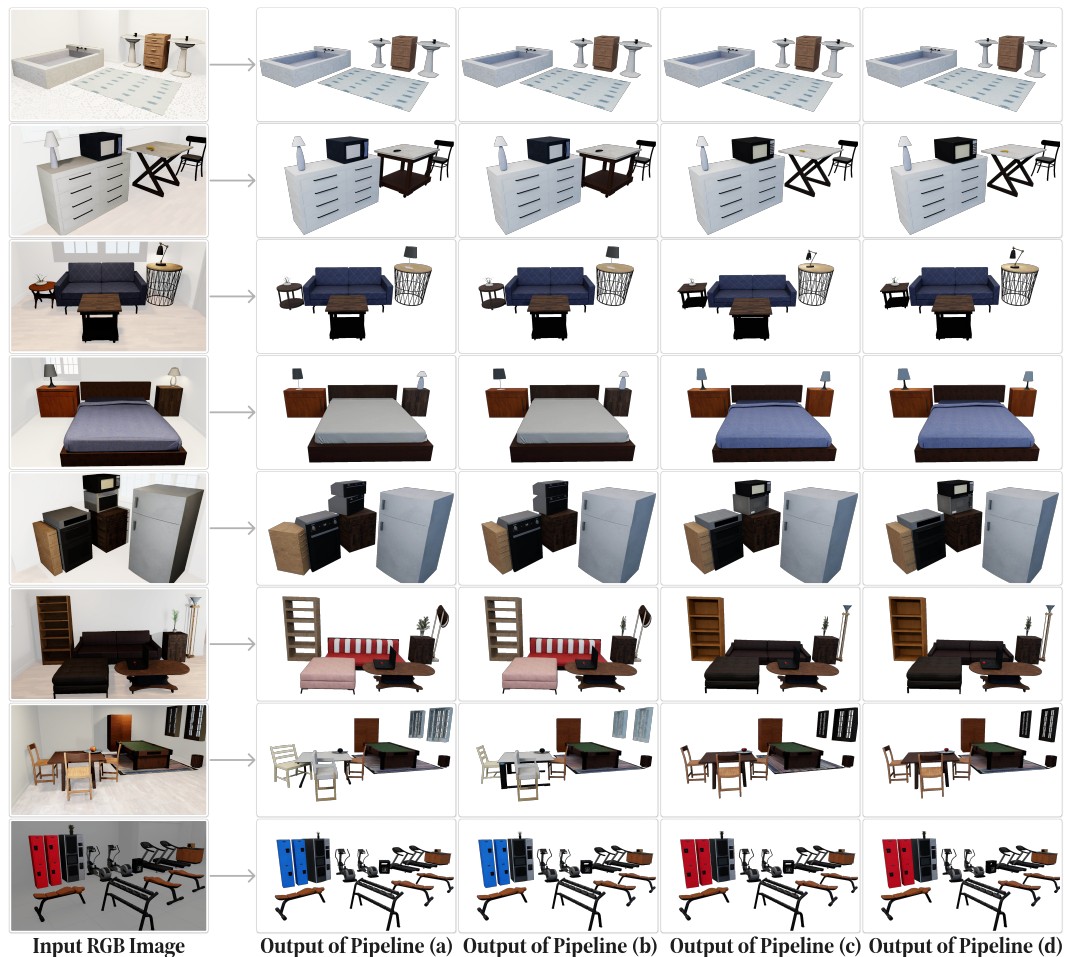

| Input RGB Image | Output of Pipeline (a) | Output of Pipeline (b) | Output of Pipeline (c) | Output of Pipeline (d) |

Figure 6: **Qualitative sim-to-sim digital cousin scene reconstruction results.** Overall, pipeline (d) gives the best scene reconstruction results, while pipeline (c) balances inference time and reconstruction quality.

generally yields reasonable results, the default scale when capturing representative model snapshots can affect the selection of the best digital cousin. To refine this process, we propose **GPT Model Selection**, which first uses DINOv2 embedding distances to select $k_{model}$ candidate models and then prompts GPT to choose the best one, with $k_{model} = 10$ in practice.

To select the best orientation $\mathbf{q}_c$ of $\mathbf{A}_c$, we first identify $k_{ori}$ candidate orientations based on DI-NOv2 embedding distances between $\mathbf{x}_i$ and all snapshots $\{\mathbf{i}_{is}\}_{s=1}^{N_{snap}}$ of the selected digital cousin $\mathbf{A}_c$. **DINO Orientation Selection** involves reorienting the asset $\mathbf{A}_c$, rescaling it, placing it in the scene as described in Section 2.1, normalizing its bounding box, and retaking a snapshot with the same relative position to the viewer camera as detailed in Appendix A.1. The best orientation $\mathbf{q}_c$ is then selected based on DINOv2 embedding distances with the retaken snapshots and $\mathbf{x}_i$. However, orientation can be defined for objects within the same category based on key features, even under different scales. For example, a taller cabinet can be considered to have the same orientation as a shorter cabinet if their frontal faces align. Motivated by this, we propose **GPT Orientation Selection**, where GPT is prompted to directly select the best orientation among the $k_{ori}$ candidate orientations, with $k_{ori} = 4$ in practice.

Table 2 presents a quantitative evaluation of our digital cousin creation in the sim-to-sim setting, while Fig. 6 provides qualitative visualizations of the output scenes for each pipeline. To ensure diversity at the object level, no model is present in more than one test scene.

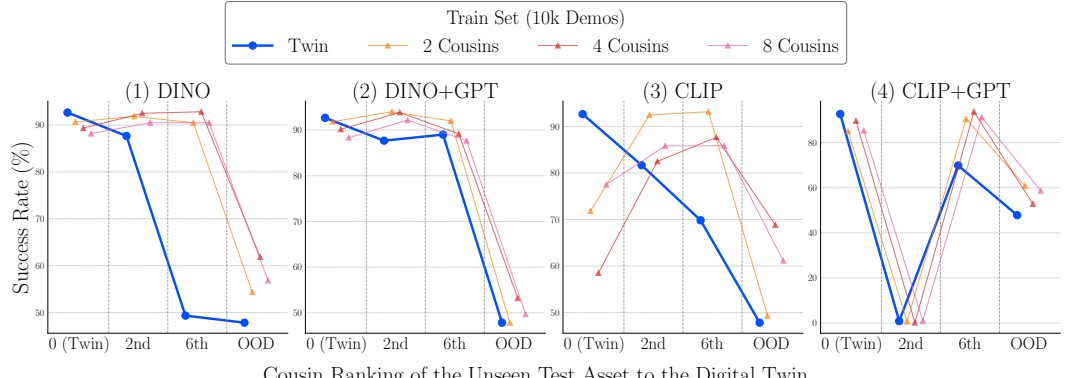

Figure 7: **Ablation study of how to choose digital cousins.** Average success rates of door opening policies trained on demonstrations collected from the exact twin, and different numbers of cousins. Policies are tested on four assets (from left to right in each line plot): the exact digital twin, the second **unseen** cousin selected by the corresponding method, the sixth **unseen** cousin selected by the corresponding method, and a more dissimilar asset (OOD), to quantify out-of-domain generalization ability.

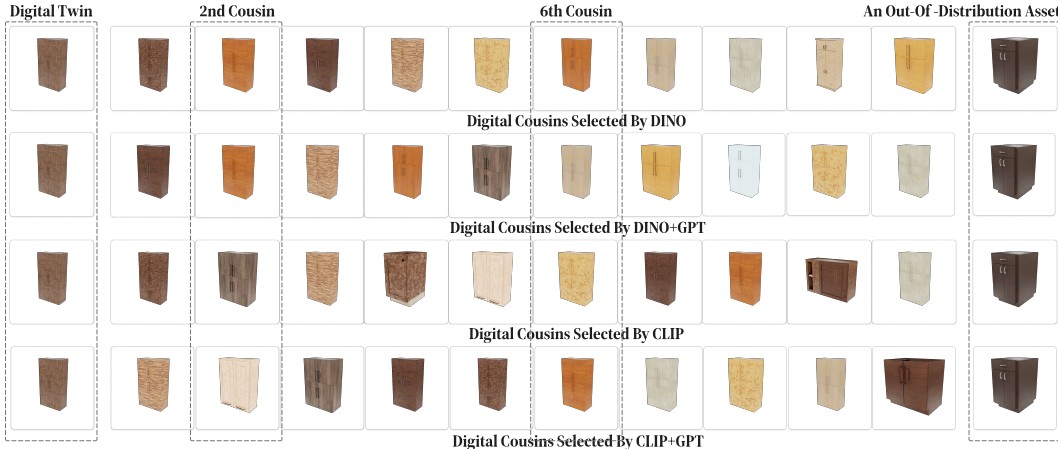

Figure 8: Visualization of digital cousins selected by different methods. Within each row, digital cousins are arranged in descending order based on their ranking. Assets enclosed in dashed boxes represent unseen test assets. DINO based methods are better than CLIP based methods for selecting geometrically similar digital cousins.

Based on the category and model matching accuracy, we observe that prompting GPT to select the nearest neighbor from a list of candidates outperforms pure DINOv2 embedding distance selection. This advantage likely stems from DINO being influenced by factors such as lighting conditions, occlusions, and changes in object scale and orientation. In contrast, GPT focuses better on geometry matching given proper prompting, which is crucial in our real-to-sim setting where an exact digital twin of an object is not always available in the simulator. Although GPT occasionally selects an incorrect model, such as the bookshelf in the sixth row of Fig. 6, it still chooses a reasonable substitute that can be appropriately scaled, oriented, and positioned to represent the target object.

Comparing (d) with (c), and (b) with (a) in terms of orientation difference and IoU-related metrics, we find that the performance of GPT Orientation Selection and DINO Orientation Selection is generally comparable. This represents a trade-off between time and robustness. Prompting GPT to select the best orientation takes less than 10 seconds per object, whereas the DINO-based method, which involves rescaling, reorienting assets, taking snapshots, and computing DINO scores, takes about 60 seconds per object but is more robust and accurate. Given that orientation will be randomized during policy training, we recommend GPT Orientation Selection for practical use. For all real-to-sim results, we adopt GPT Orientation Selection.

When comparing (b) with (d), the differences in orientation difference and IoU metrics are minimal, indicating that high-quality scenes can be reconstructed even when the assets in the simulated scene are close approximations (cousins) rather than exact replicas (twins) of the target objects.

Finally, examining the L2 Dist column in Table 2, we see that each asset is placed very close to the ground truth position. The average L2 distance errors are less than 10 cm for the first seven test scenes, and is only 17 cm for the eighth scene whose scale is 10.23 m.

We further compare DINOv2 against CLIP, another off-the-shelf visual encoder that may be used to match digital cousins. We use Door Open task to verify the best approach to match digital cousins for better policy performance. For each training set, we train policies with different hyperparameters and select the best two combinations based on the rollout success rate on the original digital twin asset. We then train policies using these best two combinations with three different seeds, resulting in six policies. The results reported in Fig. 7 are based on these six policies.

In Fig. 7, we compare four methods for selecting digital cousins:: (1) DINO: Selecting the asset with the smallest DINOv2 embedding distance to the exact digital twin; (2) DINO+GPT: First using DINOv2 embeddings to generate a candidate list, then using GPT to refine and select digital cousins from these candidates; (3) CLIP: Selecting the asset with the smallest CLIP embedding distance to the exact digital twin; (4) CLIP+GPT: First using CLIP embeddings to generate a candidate list, then using GPT to refine and select digital cousins from these candidates. The success rates of all runs used to produce Fig. 7 are shown in Table 3.

Comparing the (1)(2) with (3)(4) in Fig. 7, we can infer that DINO is a better encoder than CLIP to select digital cousins. Policies trained on demonstrations from digital cousins selected by DINO and DINO+GPT achieved approximately $90\%$ success rates on the exact digital twin and demonstrated strong generalization to the second unseen cousin. In contrast, policies trained on cousins selected by CLIP failed to exceed $80\%$ success rates on the digital twin. Interestingly, DINO+GPT appears to act as a more 'dense sampler', focusing more effectively on assets with geometric similarity to the digital twin. The observation that twin policies achieve much higher success rates on the sixth unseen digital cousin selected by DINO+GPT than the sixth unseen digital cousin select by DINO conform to this hypothesis.

Fig. 8 presents digital cousins chosen by each method. Digital cousins selected by DINO and DINO+GPT exhibit more consistent overall geometry and handle design with the digital twin than those selected by CLIP. Notably, the cousins chosen by DINO+GPT show the least geometric variance, all featuring two or four symmetrically arranged doors with similar handles to the digital twin. This observation further supports our hypothesis that DINO+GPT may serve as a more 'dense sampler' compared to DINO alone.

## B.2  Real-to-Sim Scene Generation: Additional Results

Additional qualitative results of our real-to-sim digital cousin creation and scene generation pipeline are presented in Fig. 9. For multi-view visualizations, please refer to our accompanying video and website.

Our real-to-sim digital cousin creation pipeline has the potential to create cousins and reconstruct scenes from a single RGB image without requiring ground truth camera intrinsics. We employ the Paramnet-360Cities-edina-uncentered model from PerceptiveFields [94] to estimate camera intrinsic matrix $\mathbf{K}$ from the input RGB image. Fig. 10 and Fig. 11 present the ACDC real-to-sim digital cousin scene generation results using the estimated $\mathbf{K}$. This capability may enable large-scale demonstration collection in the future by leveraging in-the-wild web images that lack ground truth camera intrinsics.

## B.3  Failure Cases

We observe that ACDC often struggles under the following conditions:

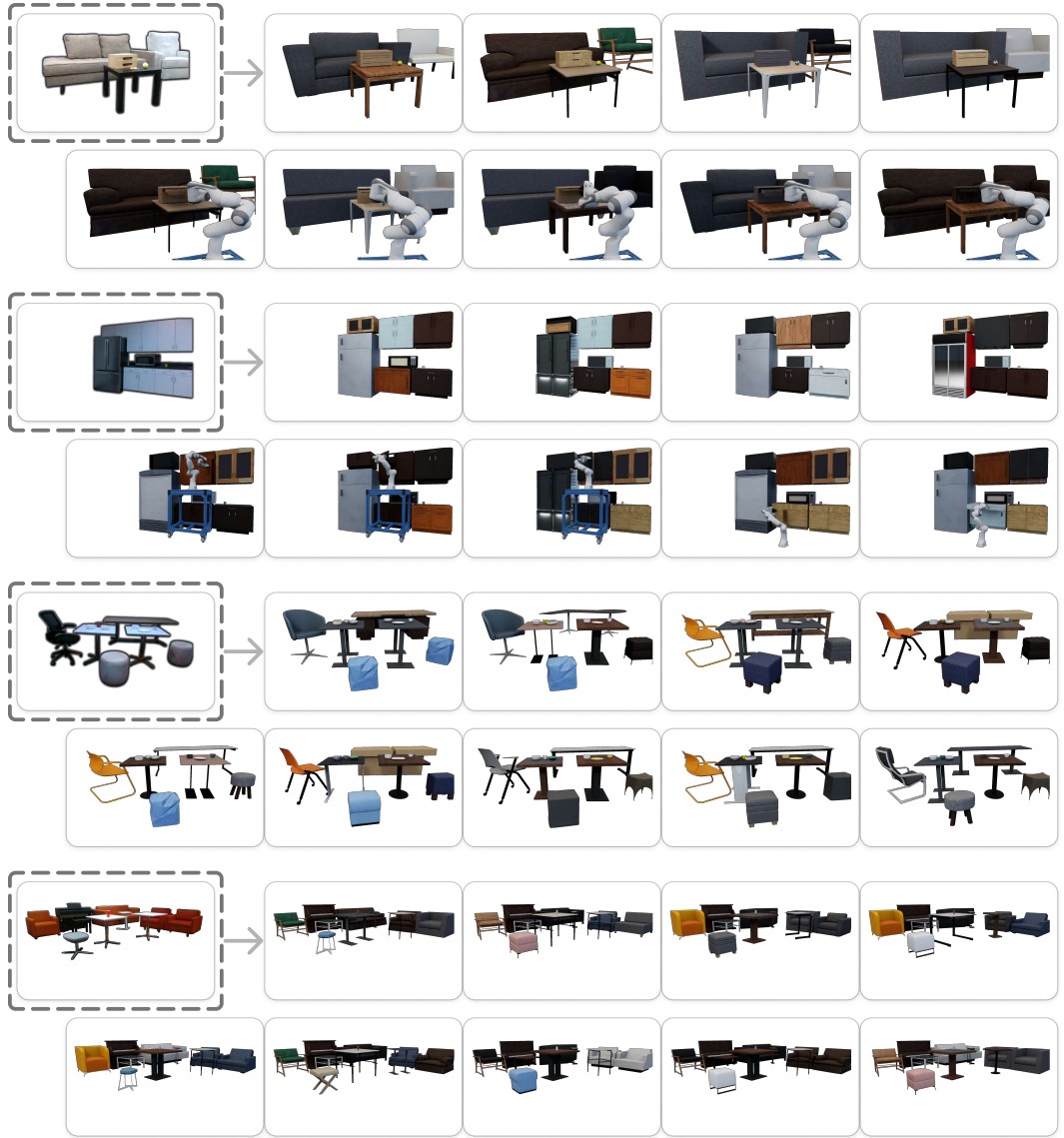

Figure 9: **Qualitative real-to-sim digital cousin scene generation results.** Multiple cousins are shown with a robot collecting demonstrations. Images cropped by dashed squares are input RGB images.

    (a). High frequency depth information

    (b). Occlusion

    (c). Semantic category discrepancies

    (d). Lack of assets within the corresponding category

    (e). Object relationships other than "on top"

The first three limitations are directly tied to how ACDC is parameterized. For (a), because ACDC relies on relatively accurate depth estimations for computing predicted object 3D-bounding boxes, poorly estimated depth maps can result in correspondingly poor object model estimations. Native depth sensors can struggle to produce accurate readings near object boundaries where discontinuities in the depth map may occur, and is compounded when an object has many fine boundaries, such as plants and fences. Moreover, because we rely on an off-the-shelf foundation model (DepthAnything-v2) to predict synthetic depth maps, we inherit its own set of limitations, such as poor predictions on

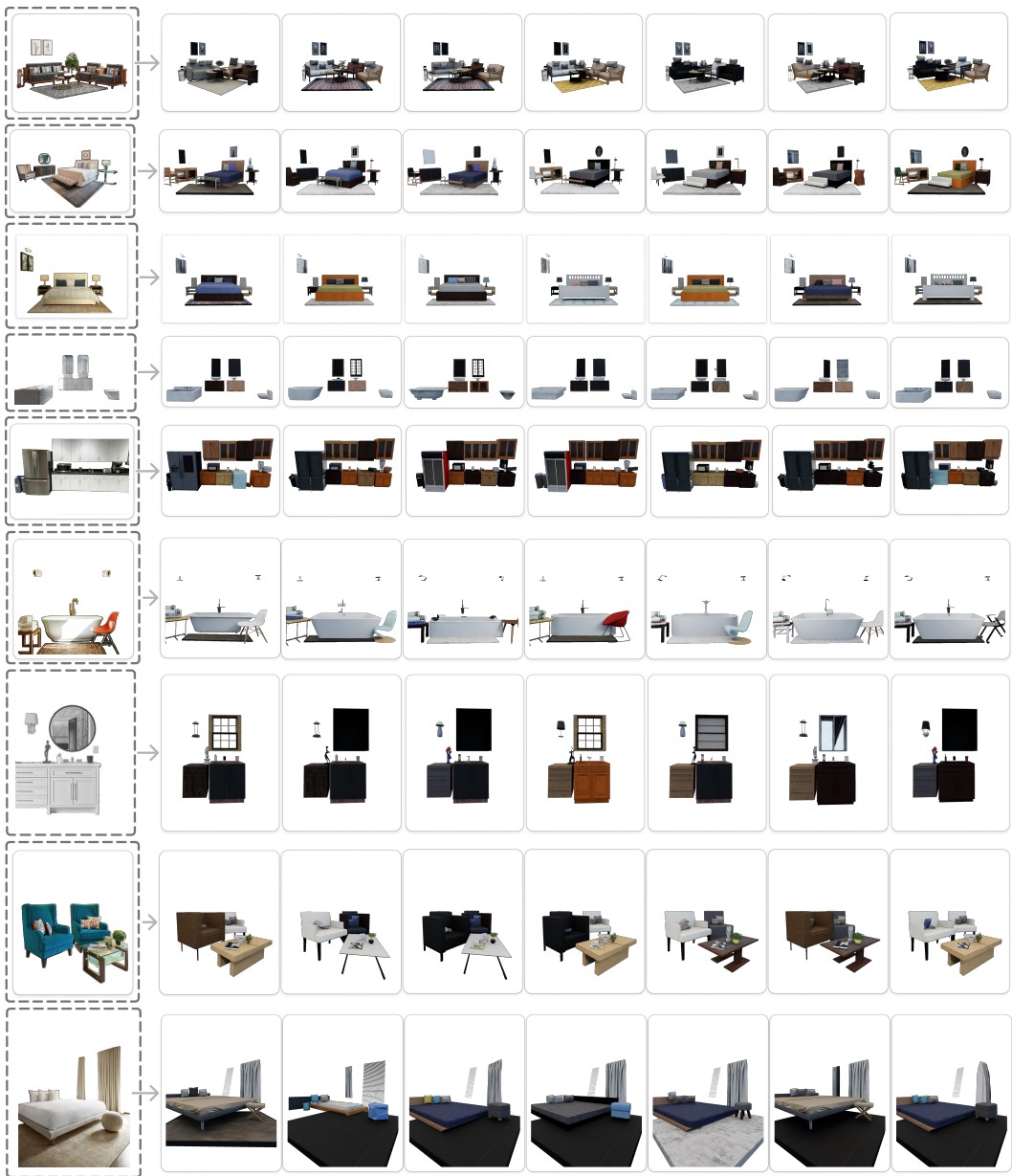

Figure 10: Qualitative real-to-sim digital cousin scene generation results **without ground truth camera intrinsics K**. Images cropped by dashed squares are input RGB images.

esoteric objects or under adversarial visual conditions. Similar to (a), occlusion (b) becomes significant when it results in an inaccurate estimation of a given object's overall bounding box. For some objects, such as cabinets and other furnitures, observing two faces is usually sufficient, but for other smooth objects, such as balls or plushes, occlusion can have nontrivial impacts on the corresponding generation of digital cousins. Lastly, ACDC can struggle when there is a mismatch between object category labels from the input RGB image and the available object asset categories from our dataset. Because we do not enforce any naming or category-abstraction level from our dataset, our category-matching method (CLIP) may fail to associate categories due to esoteric naming schemes (e.g.: bottom_cabinet_no_top) or abstraction level mismatches (e.g.: cup vs. coffee_cup vs. drinking_cup vs. water_cup), resulting in suboptimal object asset candidates when selecting digital cousins.

However, we believe that increasingly powerful foundation models can help address some of the current limitations. For instance, we have replaced DepthAnything with DepthAnything-v2 [95] ,

which offers improved depth estimation, even capturing fine-grained details more effectively. As shown in the last two rows of Fig. 11, the plant is reconstructed with greater accuracy, benefiting from the enhanced depth estimation provided by DepthAnything-v2. Using SAM-v2 instead of SAM offers better object masks. Replacing GPT-4v with GPT-4o also results in smaller orientation differences and higher bounding box IoU.

For (d), our method relies on a sufficient number of candidate assets to select digital cousins for real-world objects. This limitation can negatively impact feature matching and orientation estimation. When the number of available assets is limited within certain categories, the reconstruction quality can be sub-optimal. For instance, in BEHAVIOR-1K, there is only one pot asset, one toaster asset, and two coffee maker assets. When the input scene contains these objects, most digital cousins do not fit the corresponding category, leading to inaccurate orientation estimations due to dissimilar assets.

For (e), our method only models the "on top" relationship between objects. For other relationships, such as a kettle inside a coffee machine or books on a bookshelf, one object is placed on top of the other. However, when an object is "inside" another without a top, like a cushion in a sofa, we can still achieve reasonable reconstruction. We do this by initially placing the cushion on top of the sofa's bounding box, then moving it downward until it makes contact with the sofa.

### B.4 Comparison with URDFormer

URDFormer [31] is a recent state-of-the-art method for scene-level generation from a single RGB image, with a focus on object articulation reconstruction. As this method is quite relevant to our setup, we run a qualitative experiment to compare ACDC against URDFormer. We evaluate both ACDC and URDFormer on five real-world kitchen scenes: our kitchen scene, URDFormer's high-lighted kitchen scene, and three additional kitchen scenes. We showcase the original RGB image as well as URDFormer's and ACDC's outputs side-by-side in Fig. 12. We highlight some key differences between URDFormer and ACDC below:

- URDFormer is optimized for a trained set of object categories, while ACDC is object-agnostic and can be applied to any arbitrary set of objects.
- URDFormer can generate realistic synthetic textures from the given input image, while ACDC does not modify matched object asset textures.
- URDFormer relies on accurate bounding box information which often requires manual human annotation, whereas ACDC is fully automated and uses no human input.

In general, we find that while URDFormer can produce synthetic scene textures that visually match the real-world scene's textures, ACDC can match or even outperform URDFormer's ability to spatially reconstruct a given scene accurately, while additionally being object-agnostic (and thus able to detect and generate a much more diverse set of object categories) and fully automated with no manual human annotation.

### B.5 Policy Training Details

We train robot policies using the demonstrations collected (see Appendix A.5. Our action space is delta end-effector actions, expressed as a 6-dimensional $(dx, dy, dz)$ delta position and $(dax, day, daz)$ delta axis-angle orientation command. The commands are then executed via Inverse Kinematics (IK). Our observation space consists of {end-effector position, end-effector orientation, end-effector gripper joint state} proprioception, and a unified point cloud.

The point cloud is computed by first converting all depth images into a single point cloud with a unified frame (in our case, the robot frame), with all non-task relevant objects such as the robot and background masked out. For the real-world setting, we efficiently mask out and track all non-task relevant objects using XMem [96], allowing us to align the sim- and real-world point clouds. We then additionally add a pre-computed point cloud representation of the robot's gripper fingers,

placed at the known ground-truth location using the robot's onboard proprioception and forward kinematics. In addition to the $(x, y, z)$ per-point values, we additionally add a fourth binary value $e \in \{0, 1\}$, classifying whether that point belongs to either the scene or the robot's gripper fingers. Finally, we downsample the point cloud to a fixed size using farthest point sampling (FPS). Note that with the exception of the **Putting Away Bowl** task, the point cloud is generated from a single, over-the-shoulder camera. In the **Putting Away Bowl** task, we additionally add another over-the-shoulder camera on the other side of the robot, as well as a wrist camera, since this task exhibits much heavier occlusion during different stages compared to the other tasks.

All of our policies are trained using Behavioral Cloning with an RNN to capture the prior history of actions and a GMM to capture the distribution over demonstrations. We use a 2-layer, 512-dimension PointNet [97] encoder to encode our raw point cloud observations, which undergo further random {downsampling, translation, noise jitter} before being passed to the actor network. We also convert the binary $e$ value into a 128-dimensional learned embedding, to better enable the network to differentiate useful features between the robot fingers and the scene. Our policies use an RNN horizon of 10, RNN hidden dimension 512, are optimized using AdamW [98].

During evaluation, we take the best performing checkpoint for a given run and evaluate it 100 times. These results are then aggregated across multiple runs to give us our finalized results.

## B.6 Sim-to-Sim Policy Learning with Digital Cousins

| Task | DINO Dist. | Twin | 2 Cousins | Training Models 4 Cousins | 8 Cousins | All Assets |
|---|---|---|---|---|---|---|
| Door Opening | 0 | 96 92 91 91 87 83 | 100 96 95 92 90 74 | 95 94 94 92 87 84 | 97 95 95 94 90 88 | 94 93 86 67 66 60 |
| | 7.25 | 91 67 87 81 83 82 | 95 91 96 86 93 82 | 91 95 94 93 93 91 | 95 91 88 99 95 92 | 96 89 88 71 75 68 |
| | 7.59 | 69 60 66 65 73 72 | 98 85 93 87 95 81 | 76 91 77 95 96 87 | 91 96 98 91 87 97 | 86 83 88 65 73 74 |
| | 18.93 | 58 63 72 64 66 57 | 74 80 85 57 71 65 | 47 68 62 78 77 72 | 80 72 73 75 75 76 | 72 68 68 53 60 59 |
| Drawer Opening | 0 | 87 86 85 73 71 8 | 85 80 72 69 53 7 | 81 67 63 61 7 3 | 84 73 71 65 63 62 | 73 68 58 51 6 1 |
| | 9.42 | 87 91 89 80 85 10 | 85 86 95 93 83 10 | 92 73 75 86 8 12 | 86 78 81 82 78 72 | 73 80 79 79 12 12 |
| | 14.97 | 81 80 78 56 84 14 | 60 61 88 84 65 13 | 91 76 71 73 16 14 | 90 84 86 69 81 66 | 76 67 82 81 16 10 |
| | 17.6 | 38 36 45 30 32 17 | 37 35 41 42 13 11 | 79 32 20 23 15 8 | 97 88 94 74 90 62 | 81 75 84 80 8 15 |
| Putting Away Bowl | 0 | 33 14 14 11 9 | - | - | 11 10 9 8 5 | - |
| | 14.17 | 0 0 0 0 0 | - | - | 10 8 1 4 3 | - |
| | 14.44 | 3 0 0 0 0 | - | - | 31 14 24 31 19 | - |
| | 17.73 | 0 0 0 0 0 | - | - | 0 0 0 0 0 | - |

Table 4: Success rates (%) of all policies used in Fig. 4 and Fig. 13. "DINO Dist." shows the DINOv2 embedding distances between test assets and the original digital twin.

| Cousin Rank | Twin | Twin (↑Rand.) | Twin + All Assets(↑Rand.) | 2 Cousins | Training Models 4 Cousins | 8 Cousins | Twin + 1 Cousin | Twin + 3 Cousins | Twin + 7 Cousins | Twin+3Cousins (↑Rand.) | Twin+7Cousins (↑Rand.) |
|---|---|---|---|---|---|---|---|---|---|---|---|
| 0 | 94 93 92 | 88 88 85 | 92 86 81 | 94 93 89 | 92 92 90 | 94 91 86 | 97 94 90 | 94 91 86 | 95 94 85 | 97 96 94 | 97 93 93 |
| 2 | 85 88 90 | 90 77 76 | 91 86 89 | 94 95 93 | 92 88 95 | 91 90 88 | 91 84 89 | 96 91 96 | 86 91 92 | 93 96 94 | 92 95 99 |
| 6 | 87 91 91 | 84 81 86 | 88 90 81 | 95 92 93 | 92 86 94 | 88 88 87 | 94 93 92 | 90 90 91 | 89 93 91 | 96 98 88 | 94 94 91 |
| 11 | 2 4 2 | 36 36 26 | 41 40 49 | 39 33 47 | 25 48 34 | 58 45 44 | 7 7 24 | 36 42 47 | 40 33 29 | 40 46 42 | 35 43 45 |
| 12 | 35 49 48 | 76 69 76 | 90 82 88 | 87 96 93 | 83 85 92 | 94 92 95 | 51 52 52 | 87 82 92 | 87 88 89 | 89 93 94 | 87 97 90 |
| OOD | 62 51 38 | 50 44 54 | 76 64 76 | 43 57 46 | 43 56 63 | 51 51 64 | 60 59 54 | 53 48 51 | 69 53 32 | 70 73 70 | 55 65 65 |

Table 5: Success rates (%) of all policies used in Fig. 14. "Cousin Rank" shows the rank of test cousins selected by each method. Notice that all test assets are not seen during policy training. "OOD" stands for an asset that is not selected as top-12 digital cousin by all four methods.

As an extension of Fig. 4, Fig. 13 presents the average and standard deviations of success rates of policy rollouts on the original digital twin and multiple unseen assets. The success rates of all runs used to generate Fig. 4 and Fig. 13 are detailed in Table 4. For each training set, we train policies with different hyperparameters and select the best two combinations based on the rollout success rate on the original digital twin asset. We then train policies using these best two combinations with three different seeds, resulting in six policies. The results reported in Fig. 4, Fig. 13, and Table 4 are based on these six policies. We note that for the third **Putting Away Bowl** task, we only evaluate on five runs due to resource constraints.

An unexpected behavior is observed in the **Drawer Opening** task, where the 4-cousin policies perform sub-optimally. We believe this is due to the limited number of cabinets with drawers available for cousin selection. Among the four cousins, the first two are geometrically similar, as are the last two, but there is a significant similarity gap between the second and third cousins. This is partially illustrated by their DINO embedding distances to the digital twin: 7.78, 9.32, 14.10, and 14.90. The demonstrations collected on these four assets may not form a high-quality distribution for training. In contrast, the 4-cousin policy in the **Door Opening** task yield decent results, likely because there are more than 40 assets available for cousin selection, allowing reconstructed digital cousins to form a relatively narrower distribution. The geometric similarities between the four cousins in the **Door Opening** task are more continuous in terms of DINO similarity to the digital twin, with DINO distances being 6.49, 7.51, 8.13, and 9.66. However, 8-cousin policies still performed well in this relatively limited category, much better than all-assets policies and twin policies. A key takeaway is that: (1) when there are a sufficient number of assets to choose cousins from, all cousin policies can outperform twin policies on held-out cousins, and (2) more cousins should be found when the number of available assets is relatively small for the target category.

**Digital Cousins Improve Policy Training Stableness.** Comparing the standard deviation of policies trained on the digital twin, 8 digital cousins, and all assets in Fig. 13, we find that all-assets policies are the most unstable, followed by twin policies, while 8-cousin policies are the most stable. This highlights another advantage of training digital cousin policies: the policy training process on demonstrations collected from a set of high-quality cousins can be more stable, i.e., more robust against different random seeds and requiring less tuning.

**Digital Cousins Improve Policy Robustness.** To further examine the relative impacts of digital cousins against naive domain randomization, we re-run our sim-to-sim experiment on the **Door Opening** task against additional baselines: (a) policies trained on digital twins with increased domain randomization (greater scaling randomization: $\pm 75\%$), (b) policies trained on both the digital twin and digital cousins, where half of the dataset (5k demonstrations) are collected from the exact digital twin, and another half of the dataset (5k demonstrations) are collected from digital cousins, (c) policies trained on both the digital twin and digital cousins with increased domain randomization (greater scaling randomization: $\pm 75\%$), and (d) policies trained on both the digital twin and all assets from the nearest three categories with increased domain randomization (greater scaling randomization: $\pm 75\%$). Our results can be seen in Fig. 14. The success rates of all runs used to generate Fig. 14 are presented in Table 5. We use DINO+GPT to select digital cousins. For each training set, we train policies with different hyperparameters and select the best combination based on the rollout success rate on the original digital twin asset. We then train policies using the best combination with three different seeds, resulting in three policies. We also report policy rollout success rates on two more unseen digital cousins. Test assets are seen during training of Twin + All Assets (More Rand.) policies, but are not seen during training of other policies. Other experiment settings are the same as how Fig. 7 and Fig. 13 are produced. We find that naive domain randomization, even when increased, is insufficient to overcome the increasing domain gap when the digital twin policy is deployed on unseen cabinets. On the other hand, we find that the policies trained on the digital twin and digital cousins/all assets together exhibit similar performance compared to the policies trained exclusively on digital cousins, suggesting that perfect reconstruction via digital twins may not be necessary for sufficiently transferring a trained digital cousin policy to the original target scene.

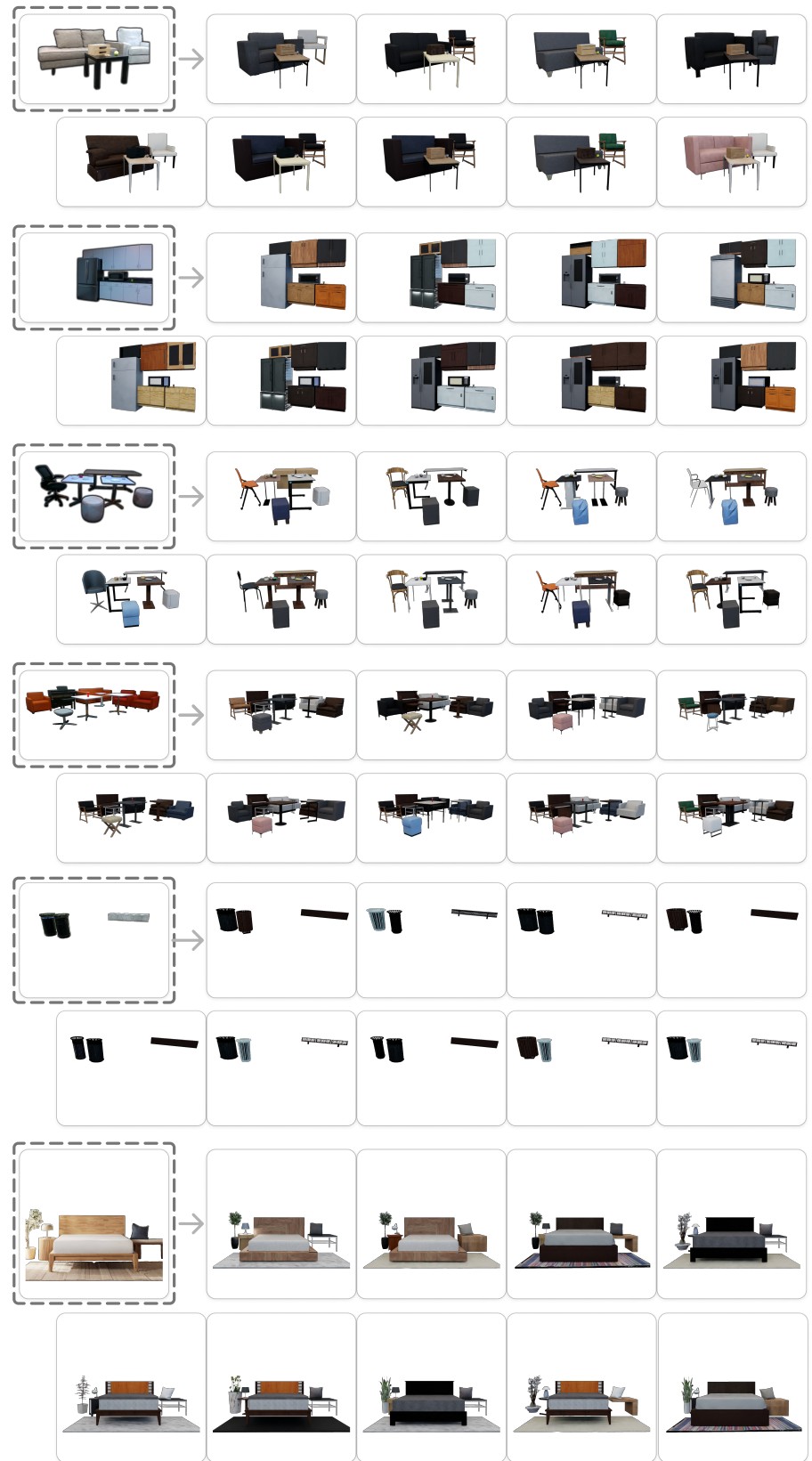

Figure 11: Qualitative real-to-sim digital cousin scene generation results **without ground truth camera intrinsics K**. Images cropped by dashed squares are input RGB images.

Input Image  URDFormer Result  ACDC Result

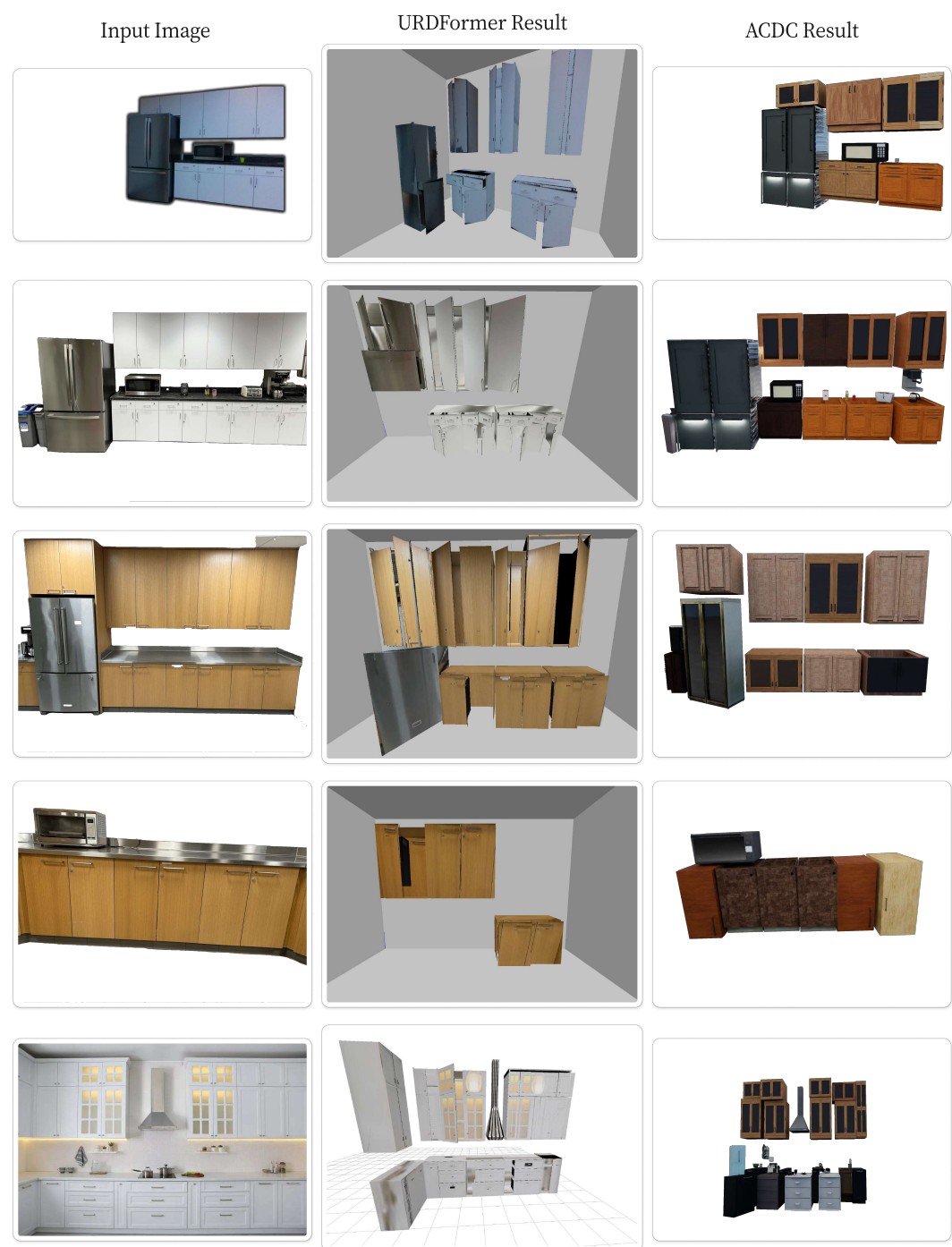

Figure 12: Qualitative comparison between ACDC and URDFormer.

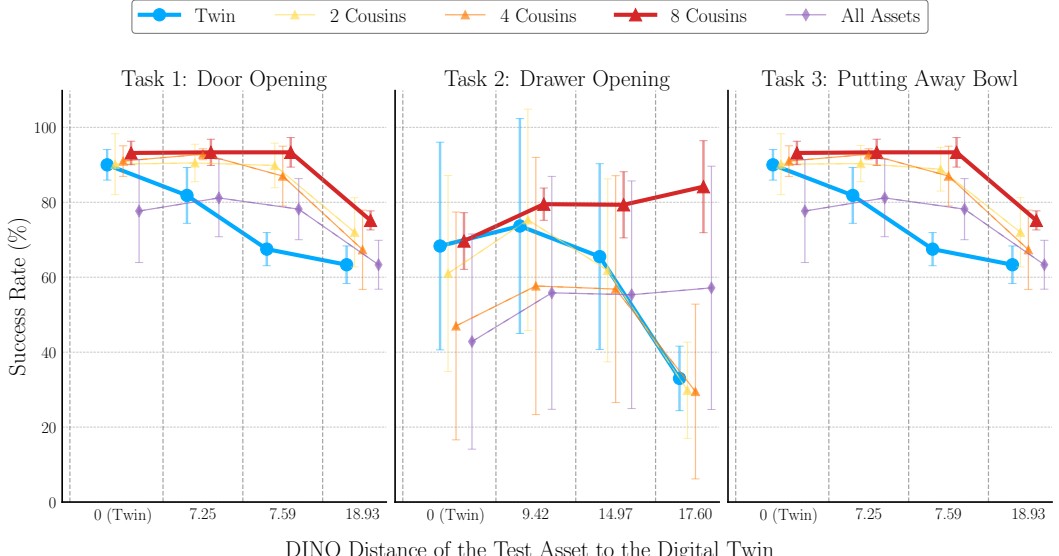

Figure 13: Average success rates (with standard deviations) of policies trained on demonstrations collected from the exact twin, different numbers of cousins, and all assets in the three nearest categories. Success rates are reported for three tasks: Door Opening, Drawer Opening, and the composite task of Putting Away Bowl. Policies are tested on four assets (from left to right in each line plot): the exact digital twin, the second unseen cousin, the sixth unseen cousin, and a more dissimilar asset, to quantify out-of-domain generalization ability. The DINO embedding distance to the digital twin is used as the quantitative metric to rank assets and select cousins. Error bars indicate the standard deviation, reflecting the stability of policy training.

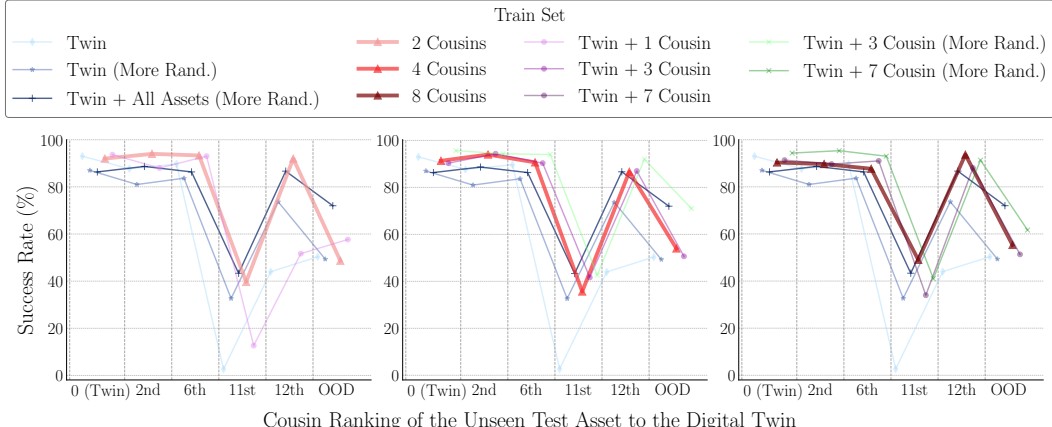

Figure 14: Average success rates of door opening policies trained on demonstrations collected from the exact twin, the exact twin with more aggressive randomization, different numbers of cousins, the exact twin with asset-level randomization, and the exact twin with asset-level randomization and more aggressive shape randomization. For Twin + Cousins and Twin + All Assets training datasets, half of the dataset is demonstrations collected from the exact twin, and another half of the dataset is demonstrations collected from different numbers of cousins or all assets from the nearest three categories. Policies are tested on six assets (from left to right in each line plot): the exact digital twin, the second **unseen** cousin, the sixth **unseen** cousin, the eleventh **unseen** cousin, the twelves **unseen** cousin, and a more dissimilar asset (OOD).

