# OpenReview forum: "Automated Creation of Digital Cousins for Robust Policy Learning"
_robot-learning.org/CoRL/2024/Conference — CoRL 2024_

### Official Review · Reviewer_hB4w · 2024-07-08
**Review of 104**

**Originality:** 3
**Technical Quality:** 4
**Clarity Of Presentation:** 3
**Potential Impact:** 3
**Recommendation:** 2
**Confidence:** 4

**Review:**

Succeses:
- The problem is important and practical.
- The authors have built an impressive system with real utility for the robotics community.
- The paper is well-written and easy to understand.
- The authors effectively use existing large-scale models (Chat-GPT, DINOv2) while providing their own useful contributions.

Areas for Improvement:
- The primary arguments seem somewhat inconsistent.  The authors claim that having many "Digital Cousins" is preferable to having a single "Digital Twin" as it provides more variety during training that can help with generalization and robustness.  While this makes sense conceptually, the variation produced by ACDC comes from the fact that the method reproduces the scene using a library of pre-existing assets, which leads to some inaccuracy between the original image and the generated scene model.  Furthermore, it seems like any variation that comes from using digital cousins could also be achieved using domain randomization on top of a more faithful digital twin to swap out different assets.  Therefore it seems like authors' argument tries to reframe inaccurate reconstruction as advantageous variation, when in reality it seems like faithful reconstruction plus controlled variation via domain randomization would probably be preferable.  It's reasonable to suggest that inaccurate digital cousins are more feasible to produce given the difficulty of generating faithful digital twins, but in that case, it would be good for the authors to focus on making that argument, and disentangle the questions of faithful reproduction from intentional variation for generalization purposes.
- Taking the above into account, the research questions seem like they could be more cleanly disentangled into: 1) How much does scene variation help with generalization and robustness when training robot policies, and 2) Can we get away with approximate models of scenes (digital cousins) instead of trying to faithfully construct exact digital twins?  The experiments in this paper effectively argue the first question, but do not satisfyingly address the second question.  In order to effectively argue the second question, it would very good to compare multiple digital cousins against a single digital twin *plus asset-swapped domain randomization*.  If the results are comparable, then this would suggest that faithful reconstruction is not necessary.  Unfortunately the existing experiments only compare against a single digital twin with no domain randomization, which only shows that variation is useful.  This is still a meaningful result, but maybe not very surprising given the long history of successes in domain randomization.  Note that I am not suggesting training on all possible assets as is discussed starting at line 196, but training on the digital twin with asset-based domain randomization designed to create similar scenes with some variation.
- There are recent efforts at automatic generation of digital twins which are not discussed here.  Several similar arguments are made in Chen et al. at RSS '24 (URDFormer: A Pipeline for Constructing Articulated Simulation Environments from Real-World Images).  It seems that this was only public a few weeks before the CoRL deadline, so direct comparison of results is not expected, however updating the text with some discussion of the relative advantages of the two systems would be very beneficial.

**Quality Of The Limitations Section:**

3

**Questions For Rebuttal:**

- Please discuss/address the issues around disentangling the questions of accurate reconstruction and intentional variation (the first bullet point under cons above).
- What are the relative benefits of ACDC compared to URDFormer?

**Robotics Focus:**

4

**Summary Of Paper:**

The paper presents a new method for building "Digital Cousins", environmental models that are similar to, but not identical copies of real world scenes from limited human data collection.

**Summary Of Recommendation:**

I have recommended weak-reject for now, but will easily improve my score if the issues discussed above are adequately addressed.  There is clearly a lot of good work here, and this is a solid contribution, but I think there are important and hopefully not too difficult adjustment that should be made before it is accepted.

---

### Official Review · Reviewer_oRdT · 2024-07-18
**Impressive method for autonomous robot learning, but some questions remain**

**Originality:** 4
**Technical Quality:** 4
**Clarity Of Presentation:** 5
**Potential Impact:** 4
**Recommendation:** 4
**Confidence:** 4

**Review:**

## Strengths:
1. This work presents a fully autonomous pipeline for constructing the simulated scene, learning a policy, and deploying it in the real world. The fact that this does not require human effort (which is required by some competing works) is impressive, and makes this method more scalable.
2. The method itself is an elegant idea. Searching across a database of 3D models for approximate matches makes sense, particularly as there are ongoing efforts to create larger and larger 3D object datasets (e.g. Objaverse). It is likely that this line of work will be impactful in the future.
3. The sim2real gap is an important problem and using digital cousins seems like a novel and sensible approach to addressing it.

## Weaknesses:
1. One of the main applications of this method (as mentioned in the paper) is a specific form of generalization to different object instances, but with the same spatial layout of the scene. For example, the robot can generalize to different instances of drawers in the same location. This is described as an advantage of digital cousins over digital twins. However, this type of generalization does not seem very useful. Why would a robot need to generalize to new cabinets in the same position? These objects do not get changed often in household environments. Generalizing to new dynamic objects such as bowls in different positions is more useful, but then there is no difference between what this digital cousins method would do and all the other works on procedurally generated simulation scenes with randomly sampled poses of objects. Therefore, this type of generalization requires much stronger motivation in the paper.
2. Given Weakness 1, the main application of the proposed method is as a domain randomization technique to bridge the sim2real gap, in case the real2sim of object models is inaccurate. Then it makes sense to train over a set of object models instead. Therefore, it should be compared against state-of-the-art techniques in sim2real domain randomization. For example, the digital twin method plus randomizing articulation joint parameters, applying random rescaling, pose randomization, applying random forces, applying random textures, etc. It is possible that some of these have already been applied in the experiments. However, they could probably be applied more aggressively, as the digital twin has a 100% success rate from 50 trials in simulation but only 25% success in real, suggesting that only mild domain randomization was applied.
3. The method for training the policy in simulation assumes access to programmed skills such as Open, Close, Pick, and Place. This may require significant engineering effort. It would be good to test how well these perform if applied directly at test time, avoiding the need for any training altogether. So, an interesting baseline could be digital twin plus directly applying these skills at test time. If this approach does not do very well, then this would motivate the use of "distillation" from scripted demos into a learned policy, as in the paper. If this baseline does perform well, then perhaps the paper could benefit from trying more difficult tasks, in order to motivate the need for simulation training.
4. The problem of goal definition in the real2sim setting is tricky but is not addressed very clearly in the paper. How does the robot know which drawer to open at test time? Is it always the top drawer? Is so, how does it know what the corresponding task is for the digital cousin objects? It would be good to discuss this further somewhere.
5. In Table 1, it is hard to tell how good this method is at precisely fitting the object models to the scene (e.g. is 0.54 Bbox IoU good for this problem?) without comparing to other methods from the computer vision community for this problem. For example, a comparison against a work like SPARC would be appropriate [2].

## Minor issues:
1. Table 1 caption: the Ori Diff metric is not very clear. What is the unit of measurement used? Is it radians?
2. The Related Work section could be more charitable to prior work on articulated object reconstruction. It would be good to be more specific about where exactly each prior work falls short. For example, in Line 242, which of these criticisms apply to “Ditto in the House”? There is also a missing paper which is highly relevant and should be discussed: Structure from Action [1].
3. Figure 4: should make clearer whether the training cousin instances do not include the “increasingly dissimilar test instance” varied along the y axis of the plots.
4. The paper says that code is provided, but I was not able to find this neither on the website nor in the supplementary materials zip file. Perhaps I missed it.
5. Line 192: “digital twins serve as oracles”. It would be helpful to make clearer that everywhere in the experiments, the digital twins are a perfect replica of the real object (if that is indeed the case).

## References:
[1] Structure from Action: Learning Interactions for Articulated Object 3D Structure Discovery (2022)

[2] SPARC: Sparse Render-and-Compare for CAD Model Alignment in a Single RGB Image (2022)

**Quality Of The Limitations Section:**

3

**Questions For Rebuttal:**

1. How long does the whole process take, from start to end (and stage by stage), for the digital cousins method vs digital twins? Specifically for the policies used to produce the results in the paper. It may be worth discussing this, because likely digital cousin training takes longer than digital twin training.
2. Figure 4: it is difficult to understand why the digital cousins method can match or even exceed the performance of the digital twin method in the zero-distance setting, where the digital twin is a perfect replica of the test-time object and there is no sim2real gap. For example, for Task 1, the method trained on just 2 cousins already matches the performance of the digital twin method. How can this be? Some more insights on this would be much appreciated. Is it because the closest cousin is a very similar object instance to the twin anyway? It may be good to see images side by side to check this. Or is it because of how the demonstrations are collected, and the distribution of robot states seen during training being wider for the digital cousin than for the digital twin?
3. Figure 4: “all assets in the three nearest categories”. Why was this baseline chosen instead of simply “all assets in the same category”? Does it make sense to define the task on an object of a different category? Comparing against all assets of the same category would help answer the question of: is there a "sweet spot" of choosing the number of cousins which are quite related to the test object but not so many that training is wasted on completely irrelevant instances.

**Robotics Focus:**

4

**Summary Of Paper:**

This paper proposes a method for creating digital cousins: simulated environments which match the scene in front of the robot at a high level (e.g. similar spatial layout and object categories), but not exactly, allowing for some variation in the specific object instances. This allows the robot to train on a variety of object instances in simulation, aiming to improve robustness and address the sim2real gap.

**Summary Of Recommendation:**

Digital cousins is an elegant method idea with potentially large impact potential, as it takes scalable simulation training and grounds it in the scene that the robot actually has in front of it, while relaxing the need for perfect digital twin reconstruction. However, there are still a few issues regarding motivation and experiment setup described above which I would like to see ironed out to make this a stronger paper. Update post-rebuttal: my concerns have largely been addressed, so I have upgraded my score to Strong Accept.

---

### Official Review · Reviewer_wrjg · 2024-07-21

**Originality:** 3
**Technical Quality:** 3
**Clarity Of Presentation:** 4
**Potential Impact:** 3
**Recommendation:** 3
**Confidence:** 4

**Review:**

Strength:
- The paper studies an important problem in robotics. Automatic simulation generation can be helpful for various robotics applications.
- The paper is overall very well written and easy to follow.
- The proposed method, although simple and straightforward, seems effective in building digital cousins.
- The experimental results are relatively comprehensive, with real-world results presented.

Weakness:
- Although there are various experiments and associated analysis, for the scene reconstruction part, the results are presented only for a few scenes: 4 in the main paper for the sim2sim setting (with additional 4 in the appendix), 3 for the real2sim setting (with additional 1 in the appendix). This is rather a small number of cases evaluated. Moreover, only qualitative results are presented for the real2sim setting.
I would encourage the author to evaluate on more scenes (e.g., 20), and report the average performance, such that the reader would have a much better understanding of how well the method works in more general cases. Showing failure cases and presenting corresponding analysis would also be very useful. Another thing is that no baseline is presented for this stage, which also rendering accessing the performance of the model a bit hard. E.g., in Table 1, there is a L2 distance of 17.09 cm for the 4th case, and a Bbox IOU of 0.55. It's hard to tell whether this is good or bad (17 cm seems fine, but 0.55 of IOU is not that high) without a baseline as comparison. I encourage the author to think of some ways to ground these numbers so a reader can more easily judge the performance. For example, if there was a baseline, another good metrics would be human preference of the generated digital cousin between the proposed method and the baseline, which might better showcase the better performance of the proposed method.

As for the baselines, I am not sure what would be the most appropriate one to compare against, but here are some that worth considering:
Yang et al, Holodeck: Language guided generation of 3d embodied ai environments., CVPR 2024.
Chen et al,  Urdformer: A pipeline for constructing articulated simulation environments from real-world images, RSS 2024
I do understand that the settings in this paper differ from the above two (e.g., holodeck generates simulation environments from a language description) and an apple-to-apple comparison may not always be feasible. However, I do think some comparison to existing methods, even modified to fit the current setting, would make the paper much more stronger.

**Quality Of The Limitations Section:**

3

**Questions For Rebuttal:**

- For the real2sim setting, how are you getting the articulation type/axis of the real-world object? E.g., a cabinet can have its hinge joint on the left edge or the right edge. However, this seems to be not considered in the current asset retrieving process.
- Line 118 - 119: "we place the asset’s bounding box center at the centroid of the corresponding input object pointcloud" -- Isn't the input point cloud in the camera frame? But I imagine you need to put objects in the simulator in the simulation's world frame? How is the conversion done here?

**Robotics Focus:**

4

**Summary Of Paper:**

This paper introduces a pipeline named ACDC that can automatically create digital cousins of a real-world scene from a single RGB image. Compared to digital twins that try to exactly replicate the real-world scene, digital cousins allow more variations in the generated scenes. The proposed pipeline first extract objects and their masks using a combination of GPT4v and grounded-SAM. It then retrieves the relevant objects and its orientation from an existing object database, by matching DINO-v2 features. Finally it places the retrieved objects into the scene. Qualitative results and some quantitative results are presented to access the quality of the generated digital cousins in both sim2sim and real2sim settings. Additionally, policy learning can be conducted in the digital cousins, and the policy learned in digital cousin is shown to perform better than the policy learned in digital twins when generalizing to more variations. The learned policy also shows zero-shot sim2real transfer to a real-world door opening task.

**Summary Of Recommendation:**

Please see the detailed review section.

---

### Author Rebuttal · Authors · 2024-08-09

Dear Reviewers and Meta-Reviewer,

We sincerely appreciate the time and effort you have dedicated to offering constructive feedback and valuable suggestions to enhance our paper. In response, we have revised the manuscript and carried out additional experiments to provide further insights and address your concerns. Below, we respond to each reviewer’s specific questions and comments. The paper has been updated with the recommended changes, which are highlighted in yellow. We are open to any further discussions.

---

### Decision · Program_Chairs · 2024-09-04

**Decision:**

Accept

**Comment:**

The reviewers thought that the method here was good and practical.

The main complaint was in the motivation of digital cousins vs digital twins + domain randomization. Reviewer hB4w states this most clearly. Instead of generating a bunch of digital cousins, why not generate a digital twin and then do domain randomization? An effective way to answer this objection would be to provide baselines using SOA digital twin methods (reviewers provided appropriate citations) and to add domain randomization. If this comparison is not appropriate, the authors will need a good argument to that effect.

It would also help to improve the experimental section. Currently there are only 3 real2sim exps and it would help to have more. It would help to have quantitative evaluations in the real2sim setting. And, the results of Table 1 are hard to interpret; the authors should try to think of different ways to evaluate or provide benchmarks that could provide context.

Post rebuttal: great work